# Progresses and Perspectives of Near-Infrared Emission Materials with “Heavy Metal-Free” Organic Compounds for Electroluminescence

**DOI:** 10.3390/polym15010098

**Published:** 2022-12-26

**Authors:** Wenjing Xiong, Cheng Zhang, Yuanyuan Fang, Mingsheng Peng, Wei Sun

**Affiliations:** 1Key Laboratory of Laser Technology and Optoelectronic Functional Materials of Hainan Province, Key Laboratory of Functional Materials and Photoelectrochemistry of Haikou, College of Chemistry and Chemical Engineering, Hainan Normal University, Haikou 571158, China; 2Xi’an Key Laboratory of Sustainable Energy Material Chemistry, MOE Key Laboratory for Non-Equilibrium Synthesis and Modulation of Condensed Matter, School of Chemistry, Xi’an Jiaotong University, Xi’an 710049, China

**Keywords:** near-infrared emitting materials, organic compounds, organic/polymer light emitting diode

## Abstract

Organic/polymer light-emitting diodes (OLEDs/PLEDs) have attracted a rising number of investigations due to their promising applications for high-resolution fullcolor displays and energy-saving solid-state lightings. Near-infrared (NIR) emitting dyes have gained increasing attention for their potential applications in electroluminescence and optical imaging in optical tele-communication platforms, sensing and medical diagnosis in recent decades. And a growing number of people focus on the “heavy metal-free” NIR electroluminescent materials to gain more design freedom with cost advantage. This review presents recent progresses in conjugated polymers and organic molecules for OLEDs/PLEDs according to their different luminous mechanism and constructing systems. The relationships between the organic fluorophores structures and electroluminescence properties are the main focus of this review. Finally, the approaches to enhance the performance of NIR OLEDs/PLEDs are described briefly. We hope that this review could provide a new perspective for NIR materials and inspire breakthroughs in fundamental research and applications.

## 1. Introduction

In 1987, the thin-film organic light-emitting diodes (OLEDs) which have high brightness and high electroluminescent efficiency at low driving voltage was launched by Tang and Vanslyke [1]. Since then, the applications of organic/polymer light emitting diodes (OLEDs/PLEDs) in solid-state lighting source and flat-panel displays have entered a new period [2,3,4]. After years of hard works, OLEDs/PLEDs have been used to achieve great breakthroughs, especially in the visible region (400–700 nm) [5,6,7,8]. Up to now, Near-infrared (NIR)-emitting organic materials have aroused growing interest on account of promising applications in some fields such as infrared signaling and displays, bio-sensing, and telecommunications. A rising number of investigations have been made to develop NIR luminescent materials with an emission wavelength longer than 700 nm due to its potential applications in bio-imaging [9,10,11,12], chemical sensors [13,14,15,16], light emitting electrochemical cells (LECs) [17,18,19,20], OLEDs [21,22,23,24] and photovoltaic cells [25,26,27,28] etc. 

To date, the materials with NIR-emitting are mainly grouped into two categories: inorganic luminescent materials including rare earth metals [29,30,31] and alkaline earth metal luminescent materials [32], and organic luminescent materials covering transition metal complexes [33,34,35,36], small molecules [37,38,39] and polymers [40,41,42]. According to the mechanism of luminescence, these emitters are divided into two types, fluorescent [43] and phosphorescent materials [44,45,46]. The luminescence is generally defined as the radiation emitted by the atoms or molecule return to the lower energy state after the material absorbing energy and jumping to the excited state of the higher energy level. The main types of luminescence include fluorescence and phosphorescence. As shown in Figure 1, the emission originated from the transition that from the lowest excited singlet state (S_1_) to the lowest excited singlet state to the singlet ground state (S_0_) is called fluorescence, while the emission from the lowest excited triplet state (T_1_) to S_0_ is called phosphorescence.

Considering that the price and rarity of the heavy metals elevate the cost and limit of their mass processing and limit the future application, a growing number of people focus on the pure organic NIR electroluminescent materials to gain more design freedom with a cost advantage. The mechanical adaptability of organic NIR light-emitting materials also makes them have broad application prospects in flexible and stretchable devices. In addition, metal-free organic light-emitting materials could be used as biocompatible substitutes for inorganic materials, and could be used in implantable, wearable or medical applications in vivo. As shown in Figure 1, in addition to traditional fluorescent materials that just could utilize the singlet excitons, there are several new types of materials that can greatly improve quantum efficiency by taking advantage of singlet and triplet excitons at the same time. (1) The materials with triplet-triplet annihilation (TTA) [47,48] can provide 62.5% energy utilization through two triplet excitons annihilating to form a higher energy triplet exciton. (2) For doublet [49,50], only one electron occupies the highest singly occupied molecular orbital (SOMO). When this electron is excited to the lowest singly unoccupied molecular orbital (SUMO), the SOMO is empty, and transition of the excited electron back to the SOMO is totally spin-allowed. (3) The hybridized local and charge-transfer state (HLCT) material [51,52,53] with the “hot excitons” which could undergo a reverse intersystem crossing process (RISC) through the high-lying channel, and then the excitons could go through a radiative transition to the low-lying locally excited (LE) state to produce a radiative exciton ratio that break through the limit of 25% of spin statistics. (4) The thermally activated delayed fluorescence (TADF) [54,55,56] materials with small singlet-triplet energy gap (ΔE_ST_) make use of efficient reverse intersystem crossing (RISC) from the lowest triplet state (T_1_) to the lowest singlet state (S_1_) so that the theoretical internal quantum efficiency can reach 100%. 

Until now, plenty of materials with high-efficiency NIR-OLEDs/PLEDs have been researched. Primitively, researchers usually utilized the rare earth metal complexes as emitters for the NIR OLEDs, such as erbium (Er) [57,58], neodymium (Nd) [59] and lanthanum (La) [60]. Nevertheless, the luminescence efficiency of these OLEDs is quite low and inefficient for practical application. Afterwards, the focus of study has turned to organic polymers, small molecules and transition metal complexes for high-efficiency NIR-OLEDs/PLEDs gradually. 

This article summarizes recent progresses on pure organic compounds and their applications in OLEDs/PLEDs. The design and development of “heavy metal-free” NIR electroluminescent materials for high-efficiency OLEDs, as well as their challenges, are also discussed.

## 2. Tuning the Emission of Materials into NIR Region

The energy gap of the organic molecules is the decisive factor in the optical and electronic properties, which means the energy separation between the highest occupied molecular orbital (HOMO) and lowest unoccupied molecular orbital (LUMO), also called the HOMO-LUMO gap (HLG) [61]. There are several approaches to construct molecules with NIR emission. Diminution of the HLG is the key to result the absorption and emission spectra. Generally speaking, when tuning the HLG of molecules, several factors should be considered, including conjugation length, bond length alternation, and donor-acceptor (D-A) charge transfer [62,63,64]. For the π-conjugated systems, extension of its conjugation length can lead to a decrease in the HLG for certain. As shown in Figure 2, Mullen et al. [65] tuned the energy gap over a quite large range from probably 1.29 eV (960 nm) to 2.15 eV (577 nm) by adjusting the number of naphthalene units of the rylenediimide dyes. And the energy gap has a gradual downtrend as the size of polycyclic aromatic hydrocarbons grow, starting from benzene to the one containing 222 carbons by the number of benzene units or sextet carbons (Figure 3) [66]. In the polycyclic aromatic hydrocarbons systems, the energy gap is generated by alternating single and double bonds. The smaller bond length alternation is, the lower energy gap of a conjugated compound will be. As a result, reducing the bond length alternation is a significant step toward the reduction of the energy gap in the conjugated systems. In addition to these, introduction of an intramolecular D-A system in organic polymers is a popular strategy of lowering the energy gap [67]. The hybridization of energy level after donor and acceptor bonding could make the energy level of HOMO higher than that of donor, while the energy level of LUMO is lower than that of acceptor, resulting in abnormally small HOMO-LUMO separation. (Figure 4a) [68,69]. For example, Skene et al. [70] synthesized azomethines compound with the maximal absorption and emission are 440 and 534 nm. In addition, on this basis, they got the push-push (D-D), pull-pull (A-A) and push-pull (D-A) azomethines by substituting the donor and acceptor groups at both ends. The D-A azomethines has the most obvious red-shifts in absorption (148 nm) and emission (126 nm), as shown in Figure 4b.

As previously mentioned, the effects on tuning the energy gap are mainly related to the individual molecules without consideration of intermolecular interaction. The intermolecular interaction, for instance, the molecular π-π stacking, hydrogen bonding and charge transfer could also have the influence on altering the band gap of the molecules in the solid states [71,72,73,74,75]. According to the above approaches, researchers have designed and organized NIR luminescent materials.

## 3. NIR Fluorescent Materials Based on Polymers

Conjugated polymers with fluorescence units have attracted a multitude of attention due to the academic and commercial value when used as the active materials in PLEDs [76]. The turn-on voltage, color purity, and stability of the devices should be optimized to accommodate PLEDs. Some of the principal advantages of conjugated polymers are easy manufacture, solution processability, low-cost, flexibility and suitability to form large area surfaces [77]. The synthetic organic flexibility is the most obvious feature of the conjugated polymers. Through the manipulation of the structures of the monomer and polymer, the physical, thermal, optical, and electrochemical properties could be adjusted for specific applications.

For conjugated polymer, the value of the energy level band gap is the key to their performance in PLEDs. The most common and effective approach is introducing D-A units consisting of various donors and acceptors to tune the HOMO-LUMO levels. In 2004, Thompson et al. [78] synthesized 2-((*E*)-2,5-bis(hexadecyloxy)-4-methylstyryl)-5-((*E*)-prop-1-en-1-yl)pyridine (**P1**) and 5-((*E*)-2,5-bis(hexadecyloxy)-4-methylstyryl)-2,3-diphenyl-8-((*E*)-prop-1-en-1-yl)pyrido[3,4-b]pyrazine (**P2**), as shown in Figure 5. The band gap of polymer lower from 2.2 to 1.8 eV when the pyridine acceptor has been replaced by the much stronger electron-accepting pyridopyrazine. And **P2** has been used in PLEDs, which show a broad emission centered in the NIR at 800 nm. The results illustrate that the incorporation of the more strongly electron-accepting unit could decrease the band gap and obtain a considerable red-shift in the photoluminescence and electroluminescence. In 2005, Gadisa et al. [79] used the same way to make the polymer bathochromic-shift. Poly(2,7-(9,9-dioctyl-fluorene)-alt-5,5-(4′,9′-di-2-thienyl-6′,7′-diphenyl-[1′,2′,5′]thiadiazolo-[3′,4′]-*g*]quinoxaline)) (**P4**) showed a significant red-shift of 245 nm in comparison with poly(2,7-(9,9-dioctyl-fluorene)-alt-5,5-(4′,7′-di-2-thienyl-2′,1′,3′-benzothiadiazole)) (**P3**) (Figure 5). In 2011, Cacialli et al. [41] reported the low-gap polymer poly[5,7-bis(5-thiophen-2-yl)-2,3-diphenyl-thieno[3,4-b]pyrazine-*alt*-2,6(4,4-bis(2-ethylhexyl))-4*H*-cyclopenta[2,1-b;3,4-b′]dithiophene] (**P5**) containing cyclopentadithiophene (CPDT) repeat units that several polymers containing it with good performance in photovoltaic applications (Figure 5) [80,81]. The band gap is only about 1.3 eV. It showed clear NIR emission at 956 nm for the unblended device with a low external quantum efficiency (*EQE*) of 0.003%. In the blended devices, an 80 nm blue-shift in the emission and an increase in the *EQE* of 0.02% can be observed. And Reynolds et al. [82] demonstrated the ability of the alternating polyfluorene (APFO) type polymer poly[9,9-dioctyl-2,7-9*H*-fluorene-*alt*-4,7-Bis-(5-bromo-3,4-dipropoxy-thiophen-2-yl)-benzo [1,2,5]thiadiazole (**P6**) employing dialkoxythiophene and benzothiadiazole (BT) units to construct donor-acceptor-donor (D-A-D) system to operate efficiently as the electroluminescent (EL) material (Figure 5). The BT moiety is frequently used in the design of the D-A type of π-conjugated organic compounds due to the low position of its LUMO energy level and good stability of BT derivatives in the reduced state [83,84,85]. The PLEDs show strong NIR electroluminescence with an emission maximum at 742 nm and a notable *EQE* of 0.3%. In 2013, Cacialli et al. [86] presented NIR-PLEDs based on a fluorene-dioctyloxy-phenylene wide-gap host material copolymerized with a low-gap emitter **P7** (Figure 5). The 1 mol% loading yielded optimum device performance with an *EQE* of 0.04% emitting at 909 nm. The high spectral purity of the PLEDs combined with their performance support the methodology of copolymerization as an effective strategy for developing NIR-PLEDs. In the next year they investigated phthalimide based polymers as new host materials due to their ambipolar and luminescent properties [87]. Using the host phthalimide-thiophene copolymer **P8** (Figure 5) combined with the bisthienyl(benzotriazolothiadiazole) emitter (D-A-D) resulted in the much more efficient single layer NIR-PLED to date with an *EQE* of 0.27% emitting at 885 nm. Then they [88] reported a new NIR emitting copolymer **P9** (Figure 5) based on 6-(2-butyloctyl)-4,8-di(thiophen-2-yl)-[1,2,3]triazolo[4′,5′:4,5]benzo[1,2-c][1,2,5]thiadiazole (TBTTT) as the NIR emitter and poly[3,3′-ditetradecyl-2,2′-bithiophene-5,5′-diyl-alt-5-(2-ethylhexyl)-4H-thieno[3,4-c]pyrrole-4,6(5H)-dione-1,3-diyl] (P2TTPD) as the host. Pure NIR emission up to 930 nm is obtained by incorporating the D-A-D segment into the P2TTPD backbone.

Another approach to lower the band gap is utilizing the polar effect caused by the heavy atoms. The notable cases of decreasing the energy gap involve the replacement of oxygen and sulfur atoms with heavier ones like selenium and tellurium in a conjugated system [71]. The band gap drop off 0.1 eV and a redshift of 90 nm in electroluminescence maximum was observed in a fluorene-based copolymers poly[2,7-(9,9-dioctylfluorene)-co-5′,5″-(4,7-diselenophen-2′-yl)-2,1,3-benzothiadiazole] (**P10**) and poly[2,7-(9,9-dioctylfluorene)-co-5′,5″-(4,7-diselenophen-2′-yl)-2,1,3-benzoselenadiazole] (**P11**) when the sulfur is replaced with the selenium (Figure 6) [40]. In 2015, Cacialli et al. [89] also showed the substitution of a sulphur atom (**P12**) with a selenium (**P13**) atom to further extend the emission in the NIR up to 1000 nm (Figure 6). PLEDs based on copolymers **P12** give the notable *EQE* (0.09%) and show an almost pure NIR EL (95% in the NIR) peaking at 895 nm.

Moreover, there are several methods to get NIR PLED using polymers as emitters. Tu et al. [90] presented the efficient D-A alternating conjugated polymer 6,7-dichloro-5-(3,3‴-dihexyl-5‴-methyl-[2,2′:5′,2″:5″,2‴-quaterthiophen]-5-yl)-8-methyl-2,3-bis(3-(octyloxy)phenyl)quinoxaline (**P14**) bearing chlorine atoms on the backbone for NIR PLEDs (Figure 6). Based on the following three points, chlorine-containing D-A alternated copolymers are expected to show special merits in PLED: (a) The covalent radius of chlorine atom is as large as 102 pm, so this steric hindrance is large enough to synthesize atropisomers [91,92]. The notable steric hindrance will not only inhibit aggregation induced emission quenching but also block nonradiative processes due to some rotational motions in the solid state; (b) chlorine-bearing D-A alternating copolymers exhibited large Stokes shift and low self-absorption; (c) the frontier orbital levels of the chlorine-bearing D-A alternating copolymers can be finely tuned by using different donor or acceptor moieties. NIR emission centered at 708 nm was obtained with brightness over 400 cd/m^2^ based on dopant/host system. Ingleson et al. [42] reported the C-H electrophilic borylation is a simple and general high yielding approach to introduce controlled concentrations of low band gap chromophores into conjugated polymer poly((9,9-dioctylfluorene)-2,7-diyl-*alt*-[4,7-bis(3-hexylthien-5-yl)-2,1,3-benzothiadiazole]-2*c*,2*cc*-diyl) (**P16**) main chains and red shift the emission by >150 nm relative to the pristine polymer poly(9,9-dioctylfluorene-alt-benzothiadiazole) (**P15**) (Figure 6). Cao et al. [93] obtained an efficient bilayer NIR-PLED with maximum *EQE* up to 2.1% (λ_max_ = 758 nm), in which where layer 1 was phenyl-substituted poly [p-phenylphenylenevinylene] derivative (P-PPV), layer 2 was 9,9-dioctylfluorene (DOF) and 4,7-bis(3-hexylthiophen)-2-yl-2,1,3-naphthothiadiazole (HDNT) copolymer 4-(4-hexyl-5-(7-methyl-9,9-dioctyl-9H-fluoren-2-yl)thiophen-2-yl)-9-(4-hexyl-5-methylthiophen-2-yl)naphtho[2,3-c][1,2,5]thiadiazole (**P17**) (Figure 6). The improvement of the diode’s performances was considered to be the irradiative excitons confined in the interface between the two layers. In 2022, Lu et al. [94] found that the thieno[3,4-b]pyrazines (TP) could be employed as acceptor and donor units simultaneously to produce a low band-gap polymer, so they utilized this unit to build three TP-based alternating π-conjugated polymers, poly[(9,9-dioctylfluorene-2,7-diyl)-(2,3-dimethylthieno[3,4-b]-pyrazine)] (**P18**), poly[(9-(1-octylnonyl)-9H-carbazole-2,7-diyl)-(2,3-dimethylthieno[3,4-b]pyrazine)] (**P19**), and poly[5,5-dioctyl-5H-dibenzo[b,d]silole-3,7-diyl)-(2,3-dimet-hylthieno[3,4-b]pyrazine)] (**P20**) (Figure 7) with the intramolecular charge transfer (ICT) characteristics. All polymers showed strong deep red/NIR emission and the emission maxima of EL devices based on the **P18**, **P19** and **P20** are 772, 764 and 775 nm, respectively, and the *EQE*_max_ is for **P20**-based PLED devices, are almost four times larger than that for **P19**-based devices. It demonstrated that silole-containing materials had excellent electronic transmission performance originating from σ*-π* conjugation. Thus, the higher device efficiency of **P20** should be related to a better balance of hole and electron currents.

In Table 1, the photophysical and electroluminescent properties of polymers from **P1** to **P20** are summarized. Although a substantial part of the PLEDs research is focused on conjugated polymer materials, their efficiencies are still far behind that of organic phosphorescent materials.

## 4. NIR Fluorescent Materials Based on Small Molecules

Due to the parity-forbidden radiative 4f-4f transitions of the rare earth ions, the corresponding LEDs usually have a nonmeasurable or very low *EQE* and low power output. In contrast, the luminescence of organic molecules originates from their allowed S_1_–S_0_ transitions and thus free from the luminescence efficiency limitation. By using phosphorescent heavy metal complexes that can effectively harvest both the singlet and triplet excitons. Unfortunately, the EL quantum efficiency drops rapidly at high current densities. Therefore, in order to develop NIR-OLEDs/PLEDs with a high *EQE*, the research on efficient and stable fluorescent NIR-emitting materials is continuing. 

Initially, attempts were made to construct NIR luminescent materials using molecules with a large area of conjugated systems. In 2006, Kageyama et al. [95] investigated that OLED (Figure 8) using tris(8-quinolinolato)aluminum (Alq_3_) highly doped with N,N′ -bis(neopentyl)-3,4:9,10-perylenebis(dicarboximide) (**M1**) as an emitting layer exhibit near-infrared EL with a peak at 805 nm originating from **M1** aggregates (Figure 9). Phthalocyanines are known to be organic semiconductors and have attracted much attention because of their high chemical stability, various synthetic modifications, epitaxial growth of thin films by organic molecular beam epitaxy and unique absorption bands extending from the ultraviolet region to infrared region [96,97]. Cheng et al. [98] reported the OLED device used purple phthalocyanine (**M2**) single crystal as an active light-emitting layer with the emission of 936 nm (Figure 9). And Du et al. fabricated NIR OLEDs based on Tetra (2-Isopropyl-5-methylphenoxyl) substituted phthalocyanine (**M3**) [99] and tetra-(methoxyphenoxy) substituted phthalocyanine (**M4**) (Figure 9) [100]. The EL intensity at about 910 nm in the devices based on **M3** was increased by about 14 times compared with the intensity at about 930 nm in the devices based on **M2** in the same device structures. The improvement in the EL intensity was attributed to the large steric hindrance of non-peripheral phenoxyl substituent of **M3**. The emission of the NIR-OLEDs based on **M4** was observed near 891 nm. Sharbati et al. [101] demonstrated an efficient NIR electroluminescence from OLED based on imine oligomer (E)-N-((E)-3-((E)-(4-iodophenyl-imino)methyl)benzyldine)-4-iodobenzenamine (**M5**) (Figure 9). Electroluminescence with peak emission wavelengths of 800 nm and maximum EQE of 1.9% were observed. Mateo-Alonso et al. [102] presented an extended and distorted member 7,8,15,16,23,24-hexaazatrianthrylene (**M6**) of the N-containing starphene family due to their excellent electron-transporting ability (Figure 9). The electroluminescence of the OLEDs based on **M6** appeared at substantially higher wavelengths (centred at 848 nm) than the previously reported heterojunctions with hexaazatrinaphtylene (HATANT) derivatives [103,104], which illustrated that electron-deficient N-containing starphenes could be considered a general platform to prepare and fine-tune the properties of NIR-OLEDs.

Among many low band gap organic compounds, the D-A type of chromophores are particularly of interest to researchers as potential NIR chromophores because their band gap levels and other properties can be readily tuned through a variety of donors and acceptors [105]. In 2008, Wang et al. [106] synthesized the NIR fluorescent compounds **M7** and **M8** with a combination of triphenylamine (TPA), thiophene, and benzo[1,2-c:4,5-c′]bis([1,2,5]thiadiazole) (BBTD) in the D-A-D system that could lead to the longest emission wavelength and the narrowest band gap (Figure 10). The TPA unit was used as donor with prominent hole-transporting ability and the BBTD-type unit was used as acceptor, which is known to possess substantial quinoidal characters within a conjugated backbone, allowing for greater electron delocalization and thus lowering of the band gap. The OLED device (**M7**) with exclusive NIR emission at 1050 nm and an *EQE* of 0.05% has been realized. By doping **M8**, the emission wavelength can be extended up to 1115 nm. One year later, Wang et al. used the same acceptor and different TPA-type donors to synthesize a series of simple NIR organic materials, 4,8-Bis[4-(N,N-diphenylamino)phenyl]benzo[1,2-c:4,5-c()bis[1,2,5]thiadiazole (**M9**); 4,8-Bis[4-(N-phenyl-N-(4-methylphenyl)amino)phenyl]benzo[1,2-c:4,5-c()bis[1,2,5]thiadiazole (**M10**); 4,8-Bis[4-(N-phenyl-N-(1-naphthyl)amino)phenyl]benzo[1,2-c:4,5-c()bis[1,2,5]thiadiazole (**M11**) and 4,8-Bis[5-(N,N-diphenylamino)-2-thiophene]benzo[1,2-c:4,5-c()bis[1,2,5]thiadiazole (**M12**) (Figure 10) [107]. The non-planar TPA unit is available to improve carrier-transporting properties and suppress aggregations. The emission-peak maxima of NIR-OLEDs based on these compounds are all above 1000 nm, and the longest EL is at 1220 nm for **M12**. Nondoped OLED (**M10**) achieved NIR emission exclusively at 1080 nm with *EQE* of 0.73%. Then Wang et al. [108] want to utilize the guest-host system with several requirements be considered and met. At first, the host materials should have high film-forming ability and carrier-transport ability. Secondly, the guest materials should have high emission efficiency. Thirdly, the emission of the host should overlap well with the absorption of guest, which is in favor of the energy transfer. Alq_3_ was choosed as the host due to its widespread application as a host for organic NIR fluorescent chromophores. A series of a D-A-D type of NIR fluorescent chromophores (4,9-Bis[4-(N,N-diphenylamino)phenyl][1,2,5]thiadiazolo-[3,4-g]quinoxaline (**M13**), 4,9-Bis[4-(N,N-diphenylamino)phenyl]-6,7-diphenyl[1,2,5]thiadiazolo[3,4-g]quinoxaline (**M14**) and 4,15-Bis[4-(N,N-diphenylamino)phenyl][1,2,5]thiadiazolo-[3,4-i]dibenzo[a,c]phenazine (**M15**)) were designed as the guest (Figure 10), which based on [1,2,5]thiadiazolo[3,4-g]quinoxaline (TQ) as an acceptor and TPA as a donor due to the D-A charge transfer absorption bands should be more suitable for the EL band of Alq_3_. The doped OLEDs emit NIR light from 748 to 870 nm with high efficiency and radiance. Particularly, the device containing **M13** as a dopant exhibits an exclusive NIR EL at 752 nm with an *EQE* of 1.12% and the largest radiance of 2880 mW Sr^−1^ m^−2^. And Xue et al. [109] reported NIR-OLEDs based on two D-A-D conjugated oligomers, 4,8-bis(2,3-dihydrothieno-[3,4-*b*][1,4]dioxin-5-yl-benzo[1,2-*c*;4,5-*c*′]bis[1,2,5]thiadiazole (**M16**) and 4,9-bis(2,3-dihydrothieno[3,4-*b*][1,4]dioxin-5-yl)-6,7-dimethyl-[1,2,5]thiadiazolo[3,4-g]quinoxaline (**M17**) (Figure 10), which had the same donor components. A maximum *EQE* of 1.6% and a maximum power efficiency of 7.0 mW/W were achieved in devices based on **M16**, whose emission peaks appeared at 692 nm. With a stronger acceptor and thus a reduced energy gap, longer wavelength NIR emissions peaked at 815 nm was achieved in **M17** based devices, although the maximum *EQE* was reduced to 0.51% due to the lower fluorescent quantum yield of the NIR emitter. Using the sensitized fluorescent device structure, the efficiencies were further increased by two to three times, leading to a maximum *EQE* of 3.1% for **M16** and 1.6% for **M17** based devices. In 2011, Reynolds et al. [110] showed a family of D-A-D oligomer **M18** (Figure 10), which used the 3,4-ethylendioxythiophene as the donor and BT as the acceptors. Introducing a functional end group tetrahydropyran (THP) onto these oligomers provided an opportunity for incorporating the π-conjugated system covalently into a more complex system, where the charge-transporting conjugated units could be used to fabricate solution-processable electrochromic devices. PLEDs based on **M18** showed the NIR emission peaked at 730 nm. Energy transfer from the matrix to the emitting oligomer can be achieved, resulting in PLEDs with pure oligomer emission. In 2012, Wang et al. [43] obtained a family of D-A-D type NIR fluorophores (4,9-Bis[4-(1,2,2-triphenylvinyl)phenyl][1,2,5]thiadiazolo-[3,4-g]-quinoxaline (**M19**), 4,9-Bis{4-[2,2-bis(4-methoxyphenyl)-1-phenylvinyl]phenyl}[1,2,5]thiadiazolo-[3,4-g]quinoxaline (**M20**), 4,8-Bis[4-(1,2,2-triphenylvinyl)phenyl]benzo[1,2-c:4,5-c′]bis-[1,2,5]thiadiazole (**M21**) and 4,8-Bis{4-[2,2-bis(4-methoxyphenyl)-1-phenylvinyl]phenyl}benzo-[1,2-c:4,5-c′]bis[1,2,5]thiadiazole (**M22**)) containing rigid nonplanar conjugated tetraphenylethene (TPE) moieties with electron-deficient [1,2,5]thiadiazolo[3,4-g]quinoxaline (QTD) or BBTD as acceptors (Figure 10). A twisted TPE had the excellent aggregation-induced emission enhancement (AIEE) and showed a higher fluorescence efficiency in the solid state than in solution [111,112,113]. So incorporation of TPE units into the chemical structures of poor fluorophores could improve their fluorescence efficiency in the solid state significantly. Non-doped OLEDs based on these fluorophores were made and exhibited EL emission spectra peaking from 706 to 864 nm. The *EQE* of these devices were ranged from 0.89% to 0.20% and remained fairly constant over a range of current density of 100–300 mA cm^−2^. The device with the highest solid fluorescence efficiency emitter **M19** showed the best performance with a maximum radiance of 2917 mW Sr^−1^ m^−2^ and *EQE* of 0.89%. In 2016, Ledwon et al. [114] synthesised a novel organic material (*E*,*E*)4,7-Bis(5-(2-(9-ethylcarbazol-3-yl)ethenyl)-4-hexylthien-2-yl)-benzo-2,1,3-thiadiazole (**M23**) with the structure D-π-A-π-D. Carbazole was utilized as the electron donor and BT as the electron acceptor unit (Figure 10). The choice of different, electron-rich and electron-poor units along the π-conjugated, organic backbone of the molecule could control the material functionality for organic electronic applications through the push-pull effect. Futhermore, the substitution pattern of carbazole modified the molecule properties [115,116,117] because carbazole has fine optical and electronic properties and high chemical stability. OLEDs based on **M23** presented efficient emission in red and infrared spectral ranges, with an *EQE* of 3.13%. Electroluminescence is not strongly affected by quenching in the solid state, which is commonly observed for other D-A compounds. In 2020, Promarak et al. [118] also designed and synthesized two NIR fluorophores **M24** and **M25** with hole-transporting, molecular bulky tercarbazole (Figure 10). The two isomeric NIR chromophores, based on thiadiazole[3,4-c]pyridine derivatives, achieved a high *Φ*_PL_ of 34% with an excellent electrical and morphological properties. The nondoped OLED (Figure 11) based on **M24** displayed NIR color emission peaked at 726 nm with high *EQE* of 1.51%, demonstrating that the bulky tercarbazole terminuses not only improved hole-transporting property, but also build in a highly steric conformation to the molecules. And then they reported [119] a new D-A-D structure type NIR emitter 4,9-Bis(3-hexyl-5-(4-(1,2,2-triphenylvinyl)phenyl)thiophen-2-yl)naphtho[2,3-c][1,2,5]thiadiazole (**M26**) bearing naphthothiadiazole and flanked with tetraphenylethene (TPE) (Figure 10), which utilized the aggregation-induced emission (AIE) as a new approach to obtain efficient NIR solid emitter. A non-doped device fabricated with **M26** emitted a bright NIR emission at 754 nm with a high *EQE* of 1.48% and high efficiency stability. Moreover, they studied two NIR fluorescent emitters 4,4′-(5′,5‴-(benzo[c][1,2,5]thiadiazole-4,7-diyl)bis(3′,4-dihexyl-[2,2′-bithiophene]-5′,5-diyl))bis(N,N-diphenylaniline) (**M27**) and 4,4′-(5″,5⁗-(benzo[c][1,2,5]thiadiazole-4,7-diyl)bis(3′,3″,4-trihexyl-[2,2′:5′,2″-terthiophene]-5″,5-diyl))bis(N,N-diphenylaniline) (**M28**) with the oligo(3-hexylthiophene) (Figure 10), which provided the tuning of the emission colour and solubility [120]. The optimized OLEDs exhibited strong emission in the NIR range (702–723 nm) with a high maximum *EQE* of 1.52% (**M27**).

In addtion to D-A-D type fluorophores, there are some other materials based on the D-A structure with NIR-emitting. In 2011, Sharbati et al. [121] showed the single layer OLEDs (Figure 12) based on conjugated A-D-A isatin Schiff 3,3′-(4,4′-oxybis (4,1-phenylene) bis (azan-1-yl-1-ylidene)) diindolin-2-one (**M29**) and 3,3′-(naphthalene-1,5-diylbis (azan-1-yl-1-ylidene)) diindolin-2-one (**M30**) (Figure 12), which exhibited a red-emitting at 640 nm with an *EQE* of 0.054% and a NIR-emitting at 700 nm with an *EQE* of 0.051%, respectively(Figure 13). 3-iminoindolinyl-2-one residue at both sides of chromophores was selected as an acceptor head, while spacer including 4,4′-phenoxybenzene and 1,5-naphthyl was designated as donor segments. Thus, the conjugated backbone between donor and acceptor segments provided the appropriate delocalization and hence could be effective in band gap lowering. They demonstrated that the absorption and electroluminescence properties were considerably affected by the conjugation length with the spectral emission peak of the device shifting from a wavelength of 630 to 700 nm in NIR region. In 2017, Liao et al. [122] presented a novel D-A-A type NIR emitter 7-(4-(di-p-tolylamino)phenyl)benzo[c][1,2,5]thiadiazole-4-carbonitrile (**M31**) comprising highly polar cyano group together with rigid benzo[c][1,2,5]thiadiazole as tandem acceptor and 4,4′-dimethyltriphenylamine as donor (Figure 12). In merits of the rational design strategy, high photoluminance quantum efficiency (PLQY) of 86%/71% in solution and in film state were successfully achieved. And excellent *EQE* of 3.8% with peak emission at 692 nm for 15% doped device and 3.1% with peak at 708 nm for nondoped device were successfully obtained. Notably, effciency roll-offs of both doped and nondoped device are flat. In the same year, Yang et al. [123] created far-red (FR)/NIR fluorophores with aggregation-enhanced emission (AEE) (Figure 15) and solution film-forming ability. They linked two butterfly-shaped 3,7-bis(alkoxyphenylbenzothiadiazol)phenothiazines by an alkyl spacer to form a pair of conjoined D-A butterflies 1,6-Bis(3,7-bis(7-(4-((2-ethylhexyl)oxy)phenyl)benzo[c][1,2,5]thiadiazol-4-yl)-10H-phenothiazin-10-yl)hexane (**M32**) as FR/NIR electroluminescent emitters (Figure 12). The non-doped solution-processed device could emit NIR light of 683 nm with an *EQE* of 0.57%, and the introduction of the hole-transporting layer PVK resulted in FR light of 659 nm with a remarkably increased *EQE* of 1.82%. In 2018, Cacialli et al. [124] incorporated a novel triazolobenzothiadiazole-based emitter 6-(2-butyloctyl)-4,8-bis(5′-(2-butyloctyl)-[2,2′-bithiophen]-5-yl)-1*H*-[1,2,3]triazolo[4′,5′:4,5]benzo[1,2-*c*][1,2,5]thiadiazole (**M33**) and a novel polymer matrix that can afford excellent spectral and transport properties (Figure 12). The EL peaked at 840 nm with maximum *EQE* of ≈ 0.15%.

Besides utilizing various donors and acceptors moiety, researchers developed a method that binding Lewis acids to a nucleophilic site in the acceptor moiety to increase its electron deficiency. This methodology was used to modulate the absorption, luminescence and charge mobility properties of D-A materials [125,126]. For example, coupling of dative bond formation with C-B bond formation, herein termed borylative fusion, would give chelated Lewis acid adducted with enhanced stability and extended π-conjugation provided by locking neighbouring aromatic units co-planar. Extended π-conjugation will further lower the LUMO energy, whilst raising the HOMO energy level, counteracting the HOMO energy level reduction normally observed on Lewis acid binding to nitrogen-atom. Thus borylative fusion represents a simple methodology for selectively modulating the LUMO energy and reducing the band gap of a material. Based on these computational study and predication, Turner et al. [127] presented a series of BT containing D-A materials with directed C-H electrophilic borylation. Solution processed OLED devices that containing 3-(9,9-dioctyl-9H-fluoren-2-yl)-12,12-dioctyl-6,6-diphenyl-6,12-dihydro-5-thia-4,5aλ^4^-diaza-6λ^4^-boraindeno[1,2-k]acephenanthrylene (**M34**) (Figure 12) as the emitter showed the maximum *EQE* values of 0.48% with a λ_max_ of 679 nm and a broad emission stretching into the NIR. In 2017, D’Ale’o et al. [128] reported a novel hemicurcuminoid boron difluoride complex (*E*)-4-(2-(2,2-difluoro-6-phenyl-2H-1λ^3^,3,2λ^4^-dioxaborinin-4-yl)vinyl)-N,N-diphenylaniline (**M35**) containing a TPA donor group, where the acetylacetonate boron difluoride group acted as a strong acceptor (Figure 12). The OLEDs that using **M35** as an emitter showed NIR emitting peak at 750 nm with an *EQE* of 0.4%. In the same year, Cacialli et al. [129] synthesized *α,β*-unfunctionalised 4,4-difluoro-4-bora-3a,4a-diaza-s-indacene (BODIPY) moieties and constructed A-D-A oligomer (*E*)-1,2-bis(5′-(5,5-difluoro-5H-4λ^4^,5λ^4^-dipyrrolo[1,2-c:2′,1′-f][1,3,2]diazaborinin-10-yl)-4-dodecyl-[2,2′-bithiophen]-5-yl)ethene (**M36**) emitting in the NIR (Figure 12). The emitting devices incorporating the **M36** NIR emitter at a relatively low concentration of 0.5% wt exhibited maximum *EQE* up to 1.1% with an EL emission peaked at 720 nm. And in 2021, Morimoto et al. [130] also investigated the NIR emission properties of BODIPY derivatives (**M37**) (Figure 12) and fabricated an NIR-emitting OLED using **M37**. The device could emit NIR light of 756 nm with an *EQE* of 1.87%. These proved that BODIPY and its derivatives were a class of potential NIR luminescent materials. 

In addition to adjust the groups and structures of fluorescent molecules, researchers have also turned their attention to the host of the OLED device, utilizing the transfer of energy between the host and the emitter to improve the efficiency. In 2018, Yang et al. [131] achieved triplet-singlet energy transfer by the Förster mechanism. The NIR-OLEDs based on N^4^,N^4^,N^9^,N^9^-tetra-p-tolylnaphtho[2,3-c][1,2,5]thiadiazole-4,9-diamine (**M38**) were optimized with a sensitizer (Figure 12), where triplet excitons could be utilized via Förster energy transfer process due to the better overlap between the sensitizer emission and the dopant absorption. As a result, the optimized device achieved an *EQE*_max_ of 0.77% with electroluminescent peak at 786 nm. In 2022, Wong et al. [132] revealed that the good spectral overlap between the emissions of exciplex co-host and the absorption of emitter (**M39**) ensured an efficient Förster resonance energy transfer for good NIR emission (Figure 12). The optimal NIR-OLED device achieved a maximum *EQE* of 5.3% with the EL peaked at 774 nm, which was the best *EQE* ever reported based on the exciplex co-host with an organic fluorescent dopant.

The photophysical and electroluminescent properties of polymers from **M1** to **M39** are summarized in Table 2. Most of the NIR organic fluorophores are D-A type or flat π-conjugated molecules. Extending conjugated system, strengthening the D-A interaction and utilizing the energy transfer from the host to emitter are the main methods to construct oligomer fluorescent materials with NIR emitting, which can also guide further design and optimization of NIR emitters for biomedical, security, and communication applications.

## 5. NIR Phosphorescent Materials Based on Small Molecules

In general, holes and electrons injected from electrodes to emitters generate excitons, and the excitons are classified into singlet and triplet excitons that are formed at a ratio of 1:3. In the case of fluorescent emitting materials, only singlet excitons can be transformed into photons, and so only 25% internal quantum efficiency (QE) is theoretically possible, where the remaining 75% of non-radiation energy is lost. Therefore, breaking spin statistics to utilize the other 75% triplet energy is the key factor to improving OLED efficiency.

As discussed previously, there are several approaches to design high efficiency materials with NIR emitting through utilizing triplet energy. One of these is to harness the triplet excitons of organic fluorescent materials involves triplet fusion (TF) [133,134,135]. The theoretical maximum singlet exciton production yield through TF is 50%, which would contribute a maximum radiative exciton ratio of up to 62.5%. To enable highly efficient NIR-OLEDs through TF, Qiao et al. [136] used the more feasible approach of efficient TF via the host rather than direct TF from the emitter, since the triplet excitons of the NIR emitter may decay dominantly via non-radiative transition according with the energy gap law. They realized high performance NIR-OLEDs via the high-efficiency TF of a bipolar host doped with a special naphthoselenadiazole emitter 4,9-bis(4-(2,2-diphenylvinyl)phenyl)naphtho[2,3-c][1,2,5]selenadiazole (**M40**) (Figure 14). Unlike typical NIR organic D-A chromophores, **M40** features a non-D-A structure and a very large HOMO/LUMO overlap, displaying strong deep-red to NIR emitting and unique ambipolar character. The corresponding photoluminescence quantum efficiency of NSeD reached 52% in solution and retained 17% in the blend film. The optimized NIR-OLEDs demonstrated a strong emission at 700 nm with a high *EQE*_max_ of 2.1% and the *EQE* remained at around 2% over a wide range of current densities from 18 to 200 mA cm^−2^. However, this method would lose some of the energy of triplet excitons, so the quantum efficiency in theory is not 100%.

For standard closed-shell organic semiconductors, holes and electrons occupy the HOMO and LUMO respectively, and recombine to form singlet or triplet excitons. The radical emitter has a SOMO in the ground state, giving an overall spin 1/2 dipole. In the high energy ground state, where both electrons and holes occupy the SOMO level, recombination returns the system to the ground state and does not emit light. However, in 2015, Li et al. [49] achieved selective hole injection into HOMO and electron injection into SOMO to form a fluorescent two-photon excited state with near unit internal quantum efficiency and proposed an open-shell organic molecule 9-(4-(bis(2,4,6-trichlorophenyl)methyl)-3,5-dichlorophenyl)-9H-carbazole (**M41**) (Figure 14) as an NIR-emitter of OLEDs. When this electron was excited to the lowest SUMO, the SOMO was empty (Figure 15). Thus, transition back of the excited electron to the SOMO is totally spin-allowed. The maximum *EQE* of the **M41**-based OLED achieved to be of 2.4% with the emission peak at 692 nm. Then Li et al. [50] optimized the structure of the devices and improved the *EQE* up to 4.3%. But this approach was very restrictive in terms of molecular design and does not have good universality. 

The hybridized local and charge-transfer excited state (HLCT) possesses two combined and compatible characteristics with a large transition moment from a local excited (LE) state and a weakly bound exciton from a charge transfer (CT) state [137,138,139]. The former contributes to a high-efficiency radiation of fluorescence, while the latter is responsible for the generation of a high fraction of singlet excitons. The twisting D-A molecule may be an ideal carrier to realize this strategy that may possess two combined and compatible characteristics with large transition moment from LE state and weakly bound exciton from CT state. Based on that, Ma et al. [140] presented a butterfly-shaped NIR D-A compound 10-hexyl-3,7-bis(7-phenylbenzo[c][1,2,5]thiadiazol-4-yl)-10H-phenothiazine (**M42**) (Figure 14) based on the “HLCT” state (Figure 16), in which phenothiazine served as the electron donor and BT as the electron acceptor. The density functional theory (DFT)-optimized ground-state geometry revealed that the phenothiazine moiety possessed a nonplanar “butterfly-like” structure with a C-S-N-C dihedral angle (θ_N_) of 142° [141]. Furthermore, the phenothiazine and BT groups are twist-linked with a distorted angle (θ_D-A_) of 145°, which is a relatively planar arrangement for D-A compounds. So **M42** also has a twisting D-A structure. The *EQE* of the undoped **M42** device is 1.54% and the emission peak is about 700 nm. In 2017, Yang et al. [142] designed a D-π-A-π-D type compound 4,4′-(naphtho[2,3-c][1,2,5]thiadiazole-4,9-diyl)bis(N,N-diphenylaniline) (**M43**) (Figure 14) with naphthothiadiazole (NZ) as acceptor and TPA as donors. The compound not only has obvious HLCT state feature but also shows aggregation-induced emission (AIE) characteristic. Attributed to its HLCT mechanism and AIE characteristic, **M43** acquires an unprecedentedly high photoluminescent quantum yield of 60% in the neat film. The optimized OLEDs achieved a maximum *EQE* of 3.9% with the emission peak at 696 nm. And Su et al. [143] reported a D-A type luminescent materials 10-phenyl-3-(9-phenylnaphtho[2,3-c][1,2,5]thiadiazol-4-yl)-10H-phenoxazine (**M44**) (Figure 14) for the applications in efficient NIR-OLEDs, which used HLCT state principle and hot exciton mechanism to achieve the compatible coexistence of high *η*_PL_. Upon a tiny linkage modification, an efficient HLCT emissive state was obtained with the greatly increased coupling between LE and CT states, leading to a significantly enhanced η_PL_ in film. Consequently, **M44** exhibited a very excellent nondoped and doped OLED performances: a nondoped NIR-OLED (*η*_EQE_ = 0.82% and *λ*_max_ = 738 nm). Ma et al. [144] showed [1,2,5]thiadiazolo[3,4-c]pyridine (PT) heterocycle as prototype acceptor unit to construct D-A materials 4-(4-(4-(diphenylamino)phenyl)-[1,2,5]thiadiazolo[3,4-c]pyridin-7-yl)benzonitrile (**M45**) (Figure 14) with the donor TPA proximal to the N-atom (pyridyl). The choice of the PT heterocycle was based on the pyridine heterocycle is more electron-deficient than BT. Therefore, a red-shift of the absorption and fluorescence could occur due to the strong CT character, resulting in low-energy emission. And the absence of steric hindrance around the nitrogen atom would result in a smaller twist angle between the PT and TPA units, which provided a more planar ground state conformation, increased overlap of the frontier molecular orbitals, ensured large oscillator strengths and facilitated the possibility of high PL efficiencies. Besides, cyano-substituent also was incorporated to further increase the electron-deficient properties of the A moiety to realize a more red-shifted emission. The NIR emission device based on **M45** achieved a maximum *EQE* of 1.47% with emission peak at 700 nm. Then Ma et al. [145] changed the acceptor to NZ with strong electron-withdrawing capability. Moreover, its unique energy level distribution was consistent with “hot exciton” characteristics [146]. Devices based 4-(9-(4-(diphenylamino)phenyl)naphtho[2,3-c][1,2,5]thiadiazol-4-yl)benzonitrile (**M46**) (Figure 14) exhibited an excellent NIR emission with EL wavelength at 702 nm and a relatively high maximum *EQE* of 1.2%. These results demonstrate that the cyanophenyl can be used as an ancillary acceptor to construct the narrow-bandgap light-emitting materials with maintenance of PL efficiency in D-A systems, especially for high-efficiency deep-red and NIR fluorescent materials. Constructing the twisting D-A molecule is the point when designing the NIR-emitting materials with HLCT states. In 2021, Miao et al. [147] synthesized three near-infrared emitters (4,4′-(6,7-dimethyl-[1,2,5]thiadiazolo[3,4-g]quinoxaline-4,9-diyl)bis(N,N-bis(4-(tert-butyl)phenyl)aniline) (**M47**), 4,4′-(6,7-dimethyl-[1,2,5]thiadiazolo[3,4-g]quinoxaline-4,9-diyl)bis(N,N-bis(4-(tert-butyl)phenyl)-3-methylaniline) (**M48**) and 4-(9-(4-(bis(4-(tert-butyl)phenyl)amino)phenyl)-6,7-dimethyl-[1,2,5]thiadiazolo[3,4-g]quinoxalin-4-yl)-N,N-bis(4-(tert-butyl)phenyl)-3-methylaniline (**M49**)) (Figure 14), of which **M49** had a more obvious HLCT feature based on the influence of the steric hindrance of the methyl substituent. Attributed to their different spatial configurations and luminescence mechanisms, the **M47**, **M48** and **M49**-doped OLEDs displayed electroluminescence with emission peaks at 728, 693 and 710 nm and maximum *EQE*s of 1.58%, 1.33% and 3.02%, respectively. It is proved that HLCT mechanism has a positive effect on luminescence efficiency. 

TADF materials have recently attracted great attention as a result of their promising applications in highly efficient OLEDs [148,149]. To achieve efficient TADF, a sufficiently small ∆*E*_ST_ between the S_1_ and T_1_ states is desired to facilitate the RISC process. Theoretically, the ∆*E_ST_* value can be minimized by controlling the separation of the HOMO and LUMO through molecular design. Therefore, TADF is often observed in intramolecular-charge-transfer (ICT) systems containing spatially separation, which meant twisted donor and acceptor moieties. In 2015, Wang et al. [150] reported the first NIR TADF molecule 7,10-bis(4-(diphenylamino)phenyl)-2,3-dicyanopyrazino phenanthrene (**M50**) (Figure 17), featuring a V-shaped D-π-A-π-D configuration with 2,3-dicyanopyrazino phenanthrene (DCPP) as the electron acceptor, diphenylamine (DPA) as the electron donor, and phenyl rings as π-conjugated bridges, which had a small ∆*E*_ST_ of 0.13 eV. DCPP was selected as the acceptor because of its large and rigid π-conjugated structure with a strong electron-withdrawing capability. DPA was used as the donor for its excellent hole-transporting capability and its steric hindrance that would diminish aggregation-caused quenching (ACQ) [151]. The OLED device employing **M50** as a nondoped emitter exhibits a maximum *EQE* of 2.1% with a 710 nm EL peak. Since then, several researchers have used DCPP and its derivatives as acceptor to construct luminescent molecules with TADF character. Chi et al. [38] synthesized a D-A-D′ type TADF molecule 7-(4-(9H-carbazol-9-yl)phenyl)-10-(4-(diphenylamino)phenyl)-dibenzo Quinoxaline-2,3-dicarbonitrile (**M51**) (Figure 17) with the DCPP as the electron acceptor, TPA and phenyl carbazole as the different donors. The asymmetric substitution with two different donors leaded to the breaking of molecular symmetry (Figure 18), which resulted in a smaller Δ*E*_ST_ and a larger promotion of SOC to afford a faster *k*_RISC_. The non-doped device exhibits a maximum *EQE* of 5.4% with an emission peak at 718 nm. And Zhao et al. [152] developed a series of TADF molecules (7,10-bis(4-(naphthalen-1-yl(phenyl)amino)phenyl)pyrazino[2,3-f][1,10]phenanthroline-2,3-dicarbonitrile (**M52**), 7,10-bis(4-(naphthalen-2-yl(phenyl)amino)phenyl)pyrazino[2,3-f][1,10]phenanthroline-2,3-dicarbonitrile (**M53**), 7,10-bis(4-(diphenylamino)phenyl)pyrazino[2,3-f][1,10]phenanthroline-2,3-dicarbonitrile (**M54**) and 7,10-bis(4-(di([1,1′-biphenyl]-4-yl)amino)phenyl)pyrazino[2,3-f][1,10]phenanthroline-2,3-dicarbonitrile (**M55**) comprised of an electron-withdrawing pyrazino[2,3-f][1,10]phenanthroline-2,3-dicarbonitrile (DCPPr) core and various electron-donating triarylamines (Figure 17). Owing to the planar and rigid DCPPr core and intramolecular H-bonding, these molecules tend to adopt horizontal dipole orientation, leading to high quantum efficiency. In consequence, superior EL performances are realized. Their non-doped OLEDs showed NIR lights (716–748 nm) with high *EQE*_max_ of 1.0–1.9%. In 2018, Wang et al. [153] developed a new dibenzo[a,c]phenazine-11,12-dicarbonitrile (DBPzDCN) acceptor for designing efficient NIR TADF emitters 3,6-Bis(4-(diphenylamino)phenyl)dibenzo[a,c]phenazine-11,12-dicarbonitrile (**M56**) by extending the π-conjugated length of the acceptor core (Figure 17). An excellent *EQE* of 7.68% with the emission peak at 698 nm was achieved in the doped NIR-OLED device based on **M56**. In 2021, Adachi et al. [154] further increased the electron-deficient by attaching four cyano groups to the acceptor and altered the D-A connection site. The NIR-TADF emitter, 11,12-bis(4-(diphenylamino)phenyl)dibenzo[a,c]phenazine-2,3,6,7-tetracarbonitrile (**M57**) (Figure 17), had a high photoluminescence quantum yield of over 40% with a peak wavelength at 729 nm. The EL peak wavelength of the **M57** based-OLED was 734 nm with an *EQE* of 13.4%. 

In 2017, Liao et al. [155] developed a wedge-shaped D-π-A-π-D emitter 3,4-bis(4-(diphenylamino)phenyl)acenaphtho[1,2-b]pyrazine-8,9-dicarbonitrile (**M58**) (Figure 19) with TADF property and a small ∆*E*_ST_ of 0.14 eV. Combining an acenaphtho[1,2-b]pyrazine-8,9-dicarbonitrile acceptor (APDC) core coupled with two DPA donor units, because the acenaphthylene based new acceptor APDC possessed stronger electron-withdrawing ability than DCPP, which was due to the trend of forming an aromatic cyclopentadienide structure with six π-electrons in central fuloranthene core [156,157]. **M58** based nondoped device exhibited NIR emission with peak wavelength of 777 nm and high *EQE* of 2.19%, and its doped device achieved excellent *EQE* of 10.19% with emission wavelength peak at 693 nm. In 2019, Qiao et al. [158] reported a novel design strategy for efficient TADF materials via J-aggregates with strong intermolecular charge transfer. To obtain the desired J-aggregates, they designed two D-A type molecules, 3-(4-(diphenylamino)phenyl)acenaphtho[1,2-b]pyrazine-8,9-dicarbonitrile (**M59**) and 3-(4-(diphenylamino)phenyl)acenaphtho[1,2-b]quinoxaline-8,9-dicarbonitrile (**M60**), based on the planar and strong electron-drawing acceptors APDC and acenaphtho[1,2-b]quinoxaline-8,9-dicarbonitrile (AQDC) (Figure 19). The two molecules were gained by futher extending the π-conjugated length of APDC, and a strong electron-donating donor TPA. The nondoped devices with neat **M59** and **M60** films exhibited strong NIR emissions with maximum *EQE*s of 3.5% (711 nm) and 5.1% (765 nm), respectively. Then Qiao et al. [159] designed a new NIR TADF emitter 3-(4-(diphenylamino)phenyl)acenaphtho[1,2-b]pyrazino[2,3-e]pyrazine-9,10-dicarbonitrile (**M61**) (Figure 19) by embedding electron-withdrawing pyrazine in the acceptor APDC to enhance intramolecular CT and realize a red-shift in emission. The doped devices with 1 wt% **M61** showed a maximum *EQE* of 1.35% with an emission peak at 722 nm, and the electroluminescence peak wavelength of the non-doped device was 1010 nm (NIR-II region) with the maximum *EQE* of 0.003%. Moreover, Bronstein et al. [160] increased the electron-deficient ability of the acceptor by attaching cyano group to APDC, so as to obtain NIR TADF molecule 4-(4-(Diphenylamino)phenyl)acenaphtho[1,2-b]pyrazine-3,8,9-tricarbonitrile (**M62**) (Figure 19) with stable ICT state. This OLED based on **M62** displayed impressive EL wavelength of over 900 nm with maximum *EQE*s of 0.019%. Fan et al. [161] also synthesized the TADF molecule 3,4-bis(4-(diphenylamino)phenyl)acenaphtho[1,2-b]quinoxaline-8,9,10-tricarbonitrile (**M63**) (Figure 19) by attaching cyano group to APQC to strengthen the electron-withdrawing capability and reduce the energy gap. The **M63**-based non-doping device showed EL peak to 910 nm with a maximum *EQE* of 0.22%. In addition to studying the acceptor, the investigators also modified the donor group. In 2021, Liao et al. [162] designed and synthesized a donor, N,N-diphenyl-9,9′-spirobi[fluorene]-2-amine (SDPA), by attaching an 9,9′-spirobifluorene (SBF) moiety into a diphenylamine block. Then these researchers incorporated SDPA and APDC to construct a NIR TADF emitter 3-(4-(9,9′-spirobi[fluorene]-3-yl(phenyl)amino)phenyl)acenaphtho[1,2-b]pyrazine-8,9-dicarbonitrile (**M64**) (Figure 19). When **M64** was applied in non-doped device, the highest *EQE* was 2.55% with a NIR emission peaked at 782 nm. The result indicate that donor modification is also an effective strategy to achieve an efficient NIR emission. 

In addition to DCPP, APDC and their derivatives, the researchers also studied other acceptor groups that applicable to D-A NIR TADF molecules. In 2017, Qiao et al. [163] utilized the TADF molecule possessing a small ∆*E*_ST_ of 0.27 eV as the sensitizing host to harvest triplet excitons in devices NIR-OLEDs doped with a special naphthoselenadiazole emitter 4,9-bis(4-(diphenylamino)phenyl)-naphtho[2,3-c][1,2,5]selenadiazole (**M65**) (Figure 20) possessing a large HOMO/LUMO overlap and displayed a strong NIR fluorescence. The optimized NIR-FOLEDs achieved high *EQE*s up to 2.65% with emission peak at 730 nm. Wang et al. [164] presented a D-π-A type TADF compound 2-(4′-(diphenylamino)biphenyl-4-yl)quinoxaline-6,7-dicarbonitrile (**M66**) (Figure 20) with strong NIR emission feature. The TPA and quinoxaline-6,7-dicarbonitrile (QCN) were employed as electron donor and acceptor, respectively. The crooked π-conjugation geometry and potential intermolecular interaction that pointed at the edge of QCN moiety may promote **M66** to adopt edge to edge aggregation and eliminate face to face or π-π stacking, which should result in the emission red-shift and relatively weaker luminescence quenching. And the large steric hindrance of TPA can certainly weaken dipole-dipole interaction and enhance the emission (Figure 21). The non-doped NIR-OLEDs achieve excellent *EQE*s up to 3.9% with emission peak at 728 nm. In 2018, Data et al. [165] synthesized a NIR TADF emitter 4,9-bis(4-(diphenylamino)phenyl)-2,7-bis(2-ethylhexyl)benzo[lmn][3,8]phenanthroline-1,3,6,8(2H,7H)-tetraone (**M67**) (Figure 20) with a naphthalene diimide core disubstituted with TPA. There is a state existing due to a more planar, conjugated geometry having a high degree of excitonic delocalization. In the planar configuration, the lowest energy triplet state of the molecule is a unique molecular state of low energy, ^3^LE_Con_, onset at 1.79 eV. So the OLED based on **M67** showed NIR emission with peak wavelength of 740 nm and *EQE* of 2.4%. Since BT groups were previously reported to be used as acceptor for D-A-D type NIR polymer [82], then Promarak et al. [166] utilized the 5,6-dicyano[2,1,3]benzothiadiazole (CNBT) as an acceptor core due to the presence of imine and multiple electron-withdrawing cyano groups. By connecting two TPA at the end of the CNBT unit, a simple D-A-D type NIR molecule 4,7-bis(4-(diphenylamino)phenyl)benzo[c][1,2,5]thiadiazole-5,6-dicarbonitrile (**M68**) (Figure 20) with a ΔE_ST_ of 0.06 eV was obtained. The electroluminescent device using **M68** exhibited an excellent performance with *EQE*_max_ of 6.57% and a peak wavelength of 712 nm. In 2022, Ge et al. [167] developed two NIR TADF fluorophores 6-(4-(tert-butyl)phenyl)-8-(4-(diphenylamino)phenyl)-6H-indolo[2,3-b]quinoxaline-2,3-dicarbonitrile (**M69**) and 6-(4-(tert-butyl)phenyl)-8-(4-(naphthalen-2-yl(phenyl)amino)phenyl)-6H-indolo[2,3-b]quinoxaline-2,3-dicarbonitrile (**M70**) by connecting N,N-diphenylnaphthalen-2-amine and TPA with a novel electron withdrawing unit 6-(4-(tert-butyl)phenyl)-6H-indolo[2,3-b]quinoxaline-2,3-dicarbonitrile (Figure 20). These emitters have the AIE effect, J-aggregate with intra- and intermolecular CN⋯⋯H-C and C-H⋯⋯π interactions, and the large center-to-center distance in the solid state can improve the emission efficiency of thin films and non-doped OLEDs. The **M69** in non-doped device exhibits the state of the *EQE*_max_ of 6.61% with EL peak located at 709 nm, and the maximum *EQE* of the **M70**-based OLED is 9.44% with EL peak at 711 nm. And Tang et al. [39] reported a NIR molecule 2-(4-cyano-8-(4-(diphenylamino)phenyl)-[1,3]dithiolo[4′,5′:4,5]benzo[1,2-c][1,2,5]thiadiazol-6-ylidene)malononitrile (**M71**) (Figure 20) with D-A structure and AIE property, which used electron-deficient dithiofluorene fused with benzothiadiazole as an acceptor. When applied in non-doped OLED devices, **M71** showed the *EQE*_max_ of 2.2% at 802 nm. In the trinary system (host, sensitizer, and emitter), an outstanding *EQE*_max_ of 14.3% peaked at 750 nm has been achieved.

In addition to adjust the ∆*E*_ST_ by enlarging the conjugated system, inserting heteroatoms or attaching cyano groups to the aromatic ring, the researchers also constructed D-A-D type NIR TADF molecules by boron bonding compounds. In 2018, Adachi et al. [168] designed a NIR TADF fluorophore ethyl 4,6-bis((*E*)-4-(diphenylamino)styryl)-2,2-difluoro-2H-1λ^3^,3,2λ^4^-dioxaborinine-5-carboxylate (**M72**) (Figure 22) using a boron difluoride curcuminoid derivative with amplified spontaneous emission (ASE) property. The strong spatial overlap between the hole wave function and the electron wave function describing the optical transition in **M72** is consistent with the large oscillation intensity, high radiative decay rate and ASE property of the molecule. The nonadiabatic coupling effect between the low altitude excited states of the D-A-D molecule is the reason why the **M72** has effective vibration and spin orbit coupling assisted TADF activity. The **M72**-based NIR OLEDs (Figure 23) with an *EQE*_max_ of nearly 10% peaked at 721 nm. And then they [169] synthesized a dimeric borondifluoride curcuminoid derivative 4,4′,4″,4‴-((1*E*,1′*E*,1″*E*,1‴*E*)-(2,2,2′,2′-tetrafluoro-2H,2′H-1λ^3^,1′λ^3^,2λ^4^,2′λ^4^-[5,5′-bi(1,3,2-dioxaborinine)]-4,4′,6,6′-tetrayl)tetrakis(ethene-2,1-diyl))tetrakis(N,N-diphenylaniline) (**M73**) (Figure 22) containing TPA donor groups and acetylacetonate borondifluoride acceptor units with ASE properties. This **M73** had an electron withdrawing group in the meso position instead of the ester function in **M72**, where the strong excitonic coupling effects occurred that should result in a substantial shift of the emission toward longer wavelengths. The most efficient OLED based on **M73** exhibited an *EQE*_max_ of 5.1% for an emission wavelength of 758 nm. In 2020, Kéna-Cohen et al. [170] modified the structure of **M72** by replacing the phenyl group of **M72** with methyl phenyl to obtain the molecule ethyl 4,6-bis((E)-4-(di-p-tolylamino)styryl)-2,2-difluoro-2H-1λ^3^,3,2λ^4^-dioxaborinine-5-carboxylate (**M74**) (Figure 22), which resulted in a bathochromic shift to longer wavelengths and improved efficiency at high doping concentrations. The **M74** was used as the dopant to demonstrate pure NIR-emitting OLEDs with an *EQE*_max_ of up to 3.8% peaked at 840 nm and a spectral full-width at half-maximum of < 40 nm. In 2022, Yang et al. [171] designed two TADF emitters (10,10′-(((4R)-1′λ^3^,4λ^4^-spiro[dinaphtho[2,1-d:1′,2′-f][1,3,2]dioxaborepine-4,2′-[1,3,2]dioxaborinine]-4′,6′-diyl)bis(4,1-phenylene))bis(9,9-dimethyl-9,10-dihydroacridine) (**M75**) and 10,10′-(((4R)-8,9,10,11,12,13,14,15-octahydro-1′λ^3^,4λ^4^-spiro[dinaphtho[2,1-d:1′,2′-f][1,3,2]dioxaborepine-4,2′-[1,3,2]dioxaborinine]-4′,6′-diyl)bis(4,1-phenylene))bis(9,9-dimethyl-9,10-dihydroacridine) (**M76**)) (Figure 22) with tetracoordinate boron geometries. The chiral group of binaphtholate and octa-hydro-binaphtholate were integrated to construct the twisted tetrahedron-like emitters. The **M75** and **M76** with the twisted configurations lead to mechanochromism, piezochromism, and AIE emission. The nondoped OLEDs devices based on **M75** and **M76** revealed the NIR emission (peaking at 716 nm and 700 nm) with an *EQE*_max_ of 1.9% and 0.7%, respectively. The red-shifted emission of devices based on **M75** is attributed to the stronger charge transfer feature of **M75**.

In Table 3, the photophysical and electroluminescent properties of polymers from **M40** to **M76** are summarized. It can be seen that organic fluorophore that can break the spin statistics to make use of both singlet and triplet excitons is a kind of promising NIR electroluminescent materials. However, the above methods have certain limitations in molecular design, which hinder the research processes and practical applications of NIR electroluminescent materials.

LECs are also a kind of solid-state lighting device, but with a single emission layer inserted between the anode and cathode, rather than a multi-layered structure made up of several layers of components like OLEDs [172]. However, most of the emitters of NIR LECs are transition metal complexes [173,174,175], only a few of heavy metal-free emitters have been reported [176]. And we have also done some research work on polymers [20]. Even utilizing the transition metal complexes with triplet exiton, the efficiency of NIR LECs is generally low (less than 1%), far behind that of OLED [177], so NIR LECs need to be further developed.

## 6. Conclusions

Shifting the spectral range of OLEDs/PLEDs from the visible to the NIR region of the electromagnetic spectrum is of great interest. To date, much efforts have been made to develop NIR phosphorescent OLEDs/PLEDs using transition metal complexes. However, high costs, limited resources of phosphorescent materials, and efficiency roll-offs at high current densities remain challenges for their applications in long-term mass production. To reduce cost and improve environmental sustainability, the development of highly efficient OLEDs/PLEDs that does not rely on heavy metal-containing compounds remains an important need. As an alternative material system, the “heavy metal-free” NIR fluorophores have been widely investigated for their cost advantage and versatility in tuning molecules. However, the *EQE* of traditional organic near-infrared fluorescent OLEDs is generally about 0.1% or even lower due to low exciton utilization rate and low fluorescence quantum yield in solid state, which has become an almost insurmountable obstacle for their further development. Therefore, several strategies have been proposed to realize high quantum efficiency in pure organic dyes by utilizing triplet energy. Nevertheless, this research field is still in its infancy, and while many examples harvesting triplet excitons are reported, only a few studies have focused on their NIR emission, particularly in terms of OLEDs/PLEDs applications. 

This review summarized the development of NIR emission materials based on organic fluorophores, and their applications for OLEDs/PLEDs. Conjugated polymers and traditional organic small molecules just use singlet excitons and present relatively low internal quantum yield. While organic fluorophores with doublet, TTA, HLCT or TADF state utilize triplet excitons directly and theoretical internal quantum yield can reach to 100%. The most remarkable is the NIR TADF material. Devices based on the molecules with TADF properties have achieved a maximum EL wavelengths of more than 1000 nm, and the OLED based on the TADF molecule has achieved a maximum *EQE* of more than 13%, so TADF channels with triplet states in the near infrared region are expected to be the next research topic. It is hoped that this review will contribute to the singularity of TADF emitter design, which will lead to practical device performance through efficient triplet utilization. 

On the other hand, these research results provide us some points to design NIR-emitting organic fluorophores like constructing twisted D-A structure rather than just enlarging π-conjugated system to minimize the ∆*E*_ST_ value, and attaching cyano groups to the aromatic ring or utilizing boron bonding units rather than just inserting heteroatoms to enhance the electron-withdrawing capability of acceptor. 

Some excellent works have been done the field of NIR OLEDs, the quantum yield of the material, the electroluminescence wavelength, stability and *EQE* of the device have been greatly improved. However, the types of high-performance organic NIR luminescence materials are still few, and the problems of low device efficiency and efficiency roll-down are still the bottlenecks in the development and application. The future directions may include the following fields: (1) develop new and efficient organic near infrared luminescence materials; (2) study the relationship between molecular structure and electroluminescence properties; (3) reveal the relationship between device properties and device processing technology, and host and guest energy levels; (4) realize the application of organic NIR luminescence materials and devices in military, optical fiber communication, biomedical imaging and other fields. 

## Figures and Tables

**Figure 1 polymers-15-00098-f001:**
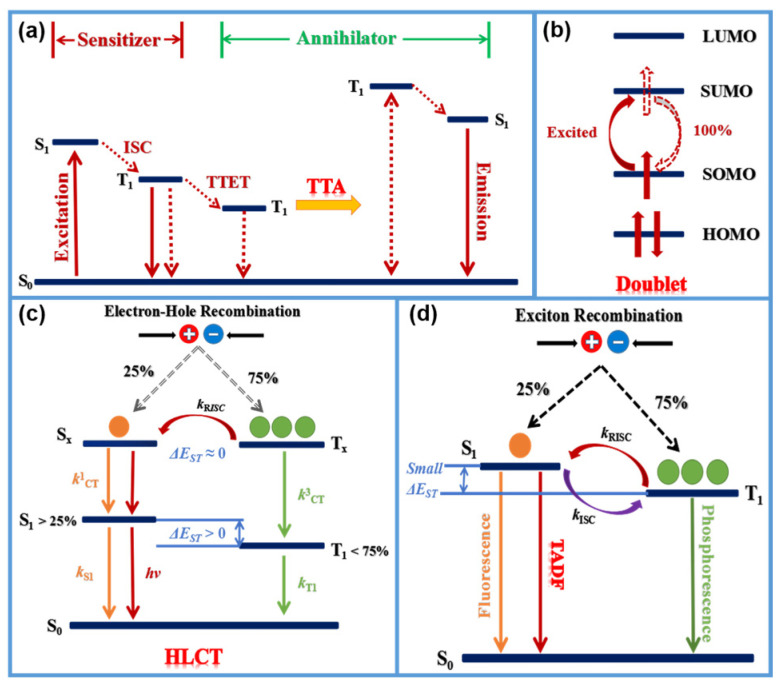
Brief mechanisms for (**a**) TTA, (**b**) doublet, (**c**) HLCT, (**d**) fluorescence, phosphorescence, and TADF.

**Figure 2 polymers-15-00098-f002:**
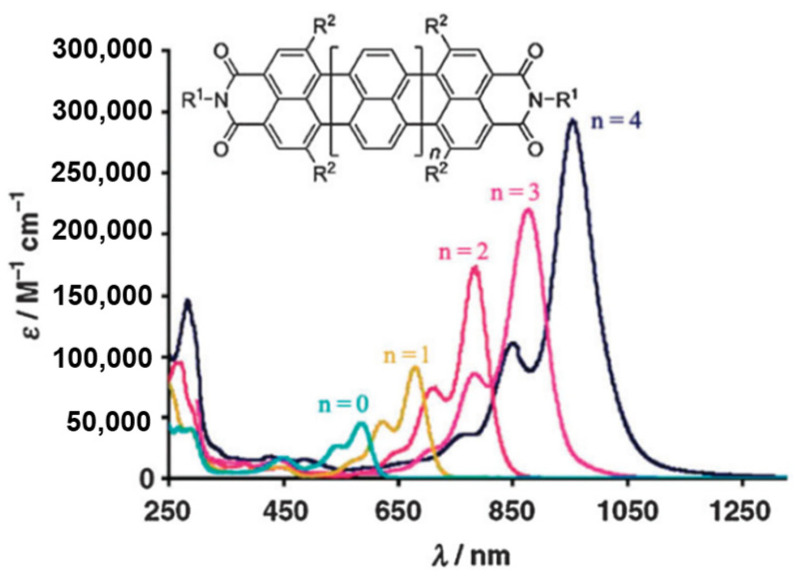
Absorption spectra of the entire tetraphenoxy-substituted rylenediimide series in CHCl_3_ (adapted from ref. [65]).

**Figure 3 polymers-15-00098-f003:**
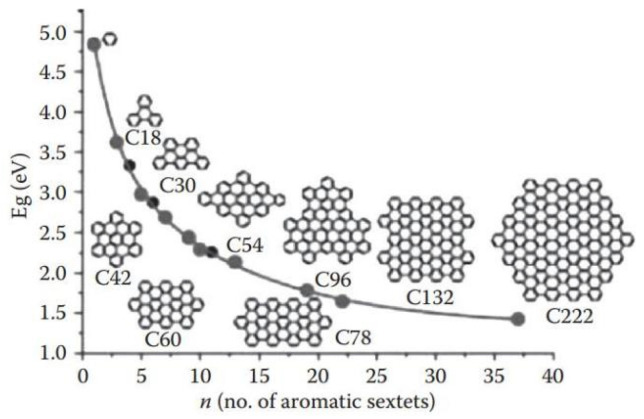
Correlation of the energy gap versus the number of aromatic sextets. (adapted from ref. [66]).

**Figure 4 polymers-15-00098-f004:**
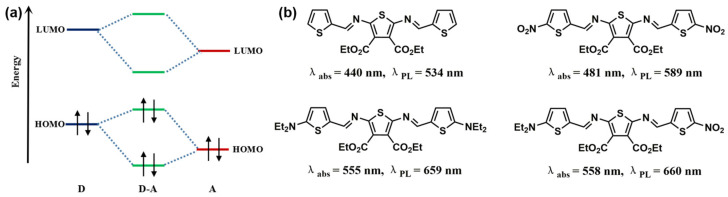
(**a**) Interaction of energy levels of a donor (D) and acceptor (A) leading to a narrower HLG; (**b**) Chemical structures of push-push (D-D), pull-pull (A-A) and push-pull (D-A) azomethines.

**Figure 5 polymers-15-00098-f005:**
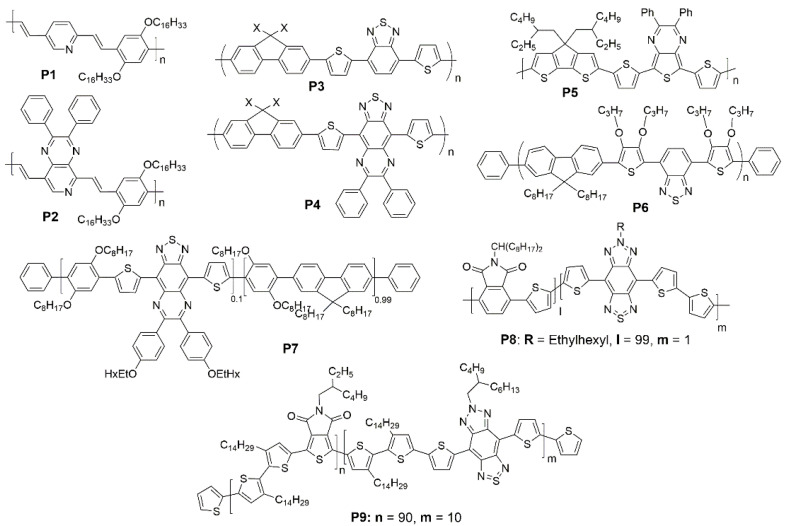
Chemical structures of **P1** to **P9**.

**Figure 6 polymers-15-00098-f006:**
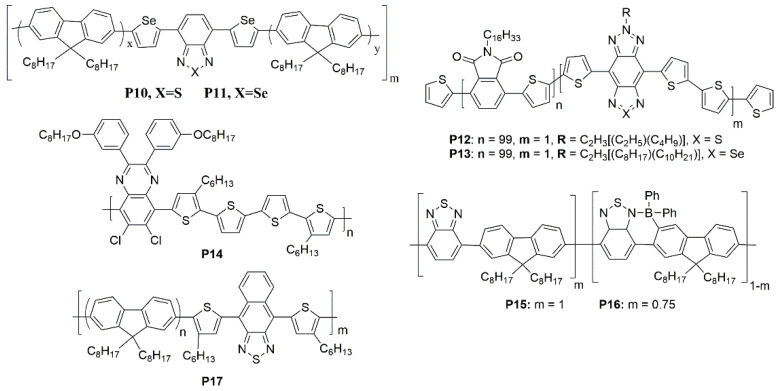
Chemical structures of **P10** to **P17**.

**Figure 7 polymers-15-00098-f007:**
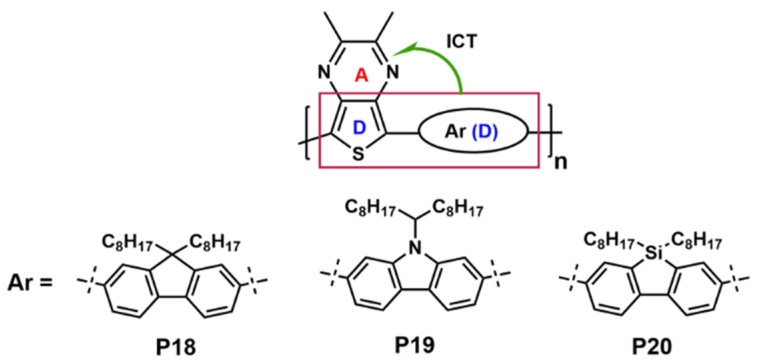
ICT interaction of **P18** to **P20**.

**Figure 8 polymers-15-00098-f008:**
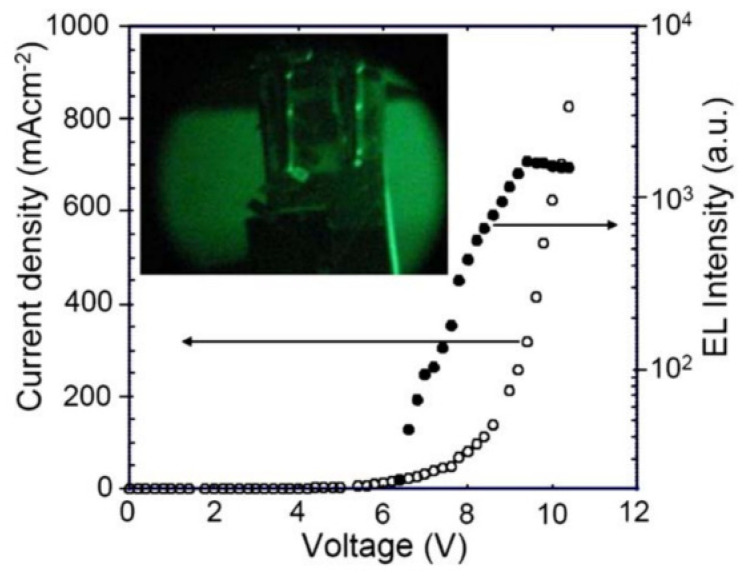
Current density–voltage-luminance characteristics of the **M1**-based OLED device. The upper inset shows a photo of near-infrared EL obtained for the device at a drive voltage of 9.4 V, taken with an infrared scope (adapted from ref. [95]).

**Figure 9 polymers-15-00098-f009:**
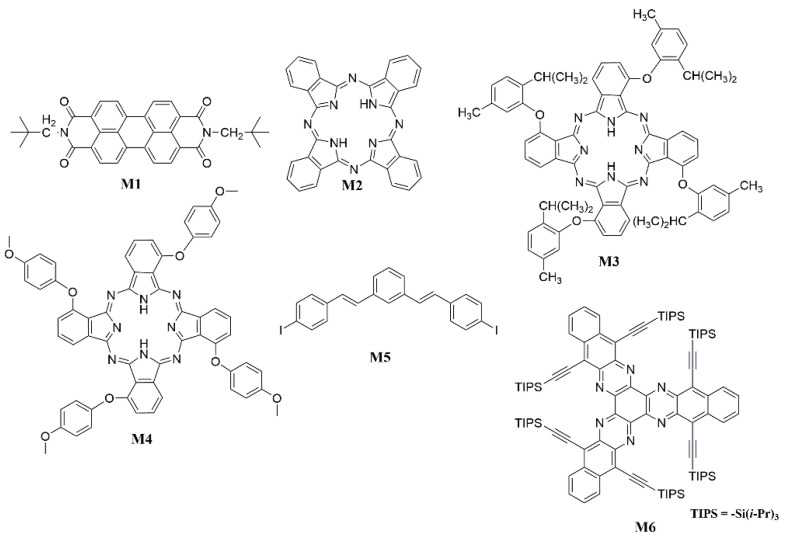
Chemical structures of **M1** to **M6**.

**Figure 10 polymers-15-00098-f010:**
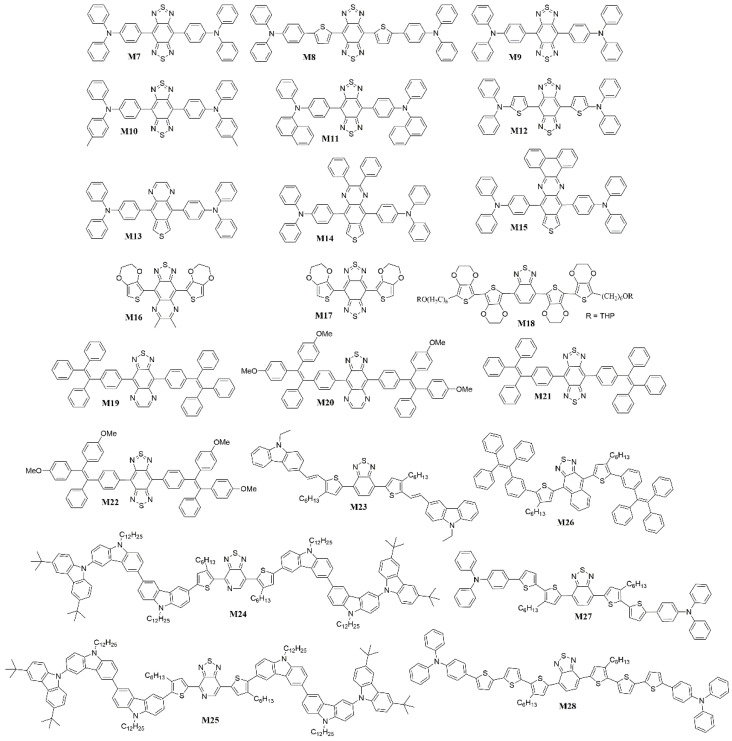
Chemical structures of **M7** to **M28**.

**Figure 11 polymers-15-00098-f011:**
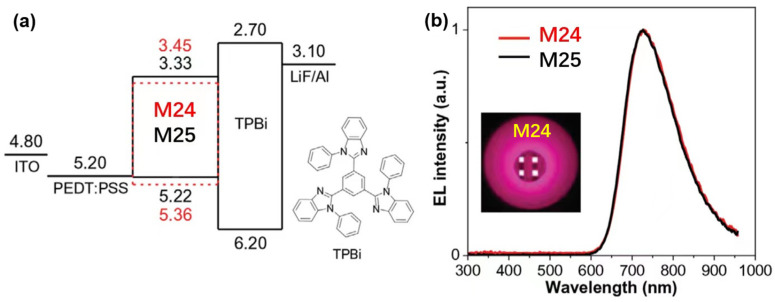
(**a**) Schematic energy band diagrams of EL devices using **M24** and **M25** as nondoped EML; (**b**) Compared EL spectra of the OLEDs based on **M24** and **M25**. Insert: photograph of **M24**-based device (adapted from ref. [118]).

**Figure 12 polymers-15-00098-f012:**
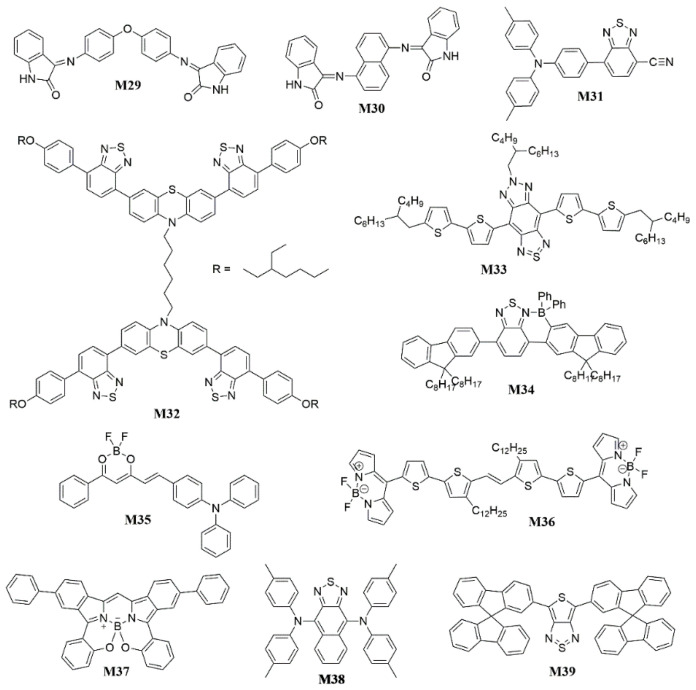
Chemical structures of **M29** to **M39**.

**Figure 13 polymers-15-00098-f013:**
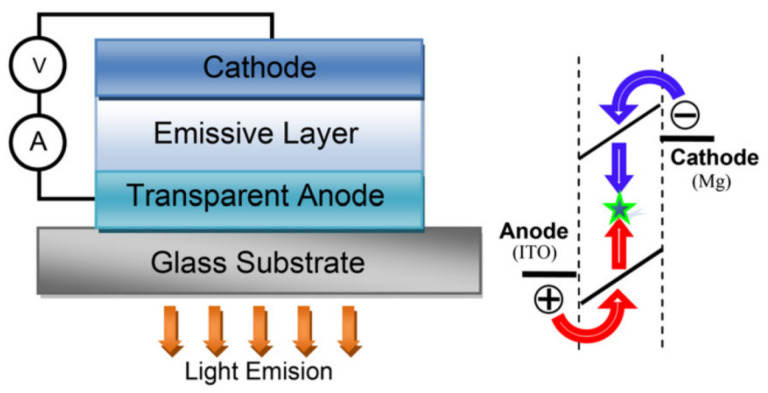
Structure of a single layer OLED based on **M29** and **M30**. Holes and electrons are recombined in the organic layer that generates light. Cathode was Mg and transparent anode is ITO (adapted from ref. [121]).

**Figure 14 polymers-15-00098-f014:**
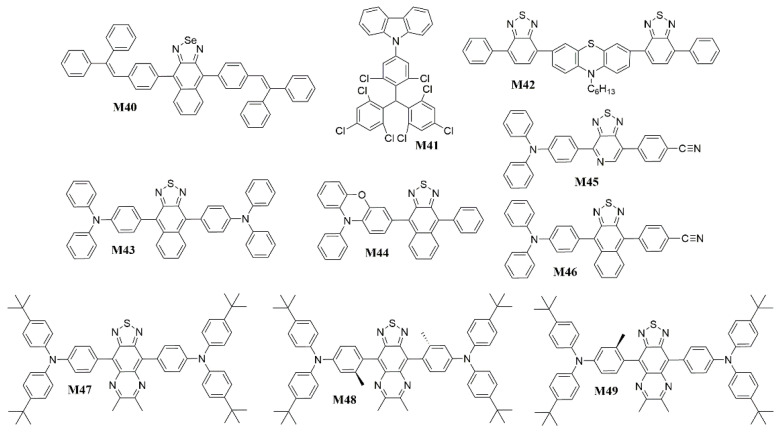
Chemical structures of **M40** to **M49**.

**Figure 15 polymers-15-00098-f015:**
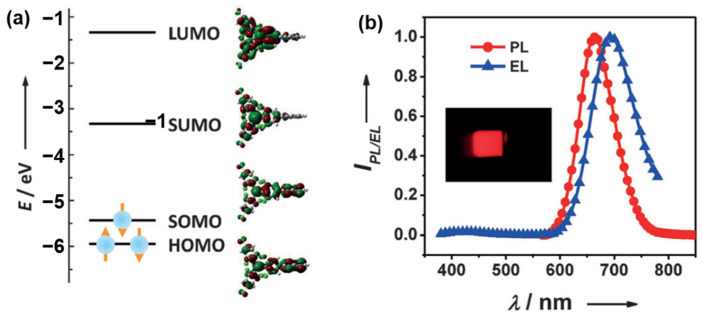
(**a**) The energy levels (left panels) and contour plots (right panels) of the molecular orbitals (LUMO, SUMO, SOMO, and HOMO) of **M41**. (**b**) The EL spectrum (7 V) of the OLEDs accompanied by the PL spectra of the doped thin film. The inset shows a photograph of the **M41**-based OLEDs operating at 7 V (adapted from ref. [49]).

**Figure 16 polymers-15-00098-f016:**
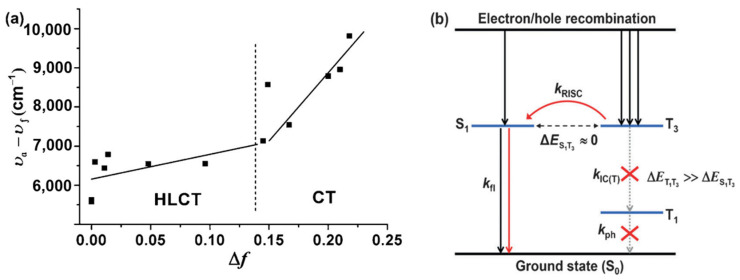
(**a**) Linear correlation of the orientation polarization (∆*f*) of solvent media with the Stokes shift (*v*_a_-*v*_f_; a: absorbed light; f: fluorescence) for **M42**. (**b**) Model for exciton relaxation in the EL process. RISC: reverse intersystem crossing; IC(T): internal conversion between the triplet states; fl: fluorescence; ph: phosphorescence (adapted from ref. [140]).

**Figure 17 polymers-15-00098-f017:**
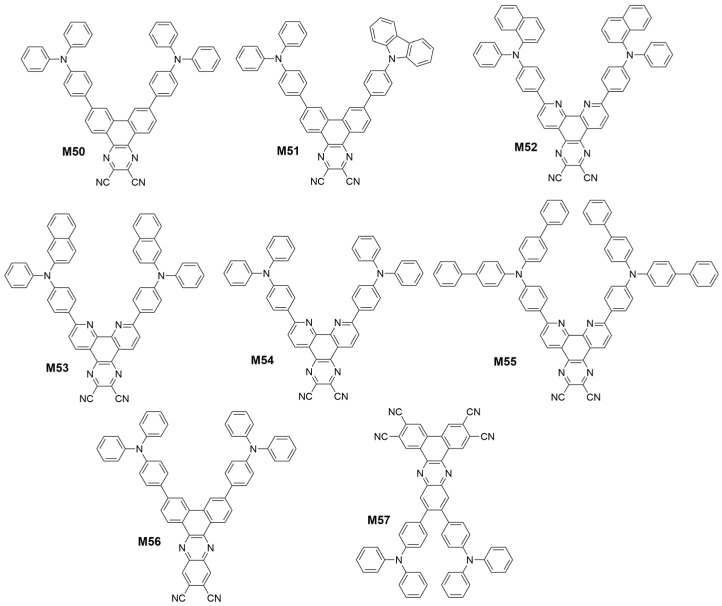
Chemical structures of **M50** to **M57**.

**Figure 18 polymers-15-00098-f018:**
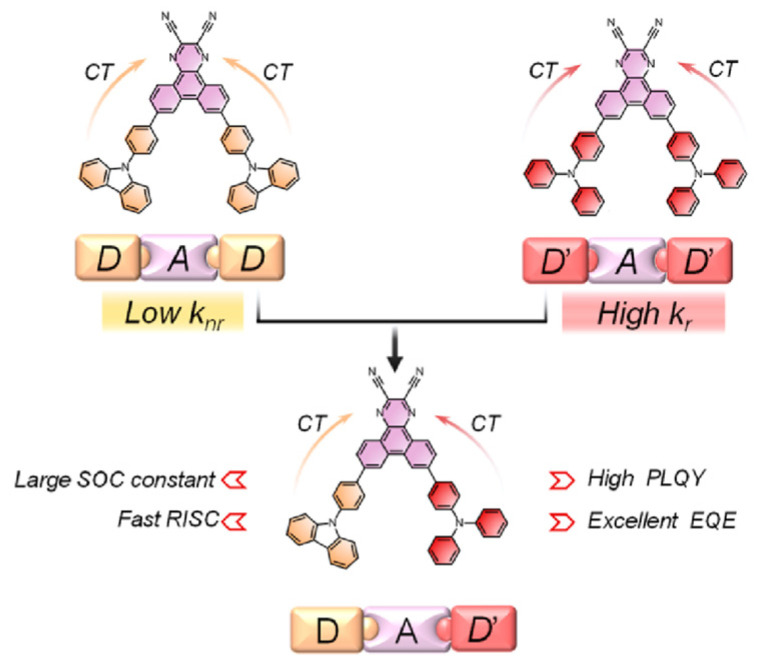
Molecular design strategy and chemical structures of **M51** (adapted from ref. [38]).

**Figure 19 polymers-15-00098-f019:**
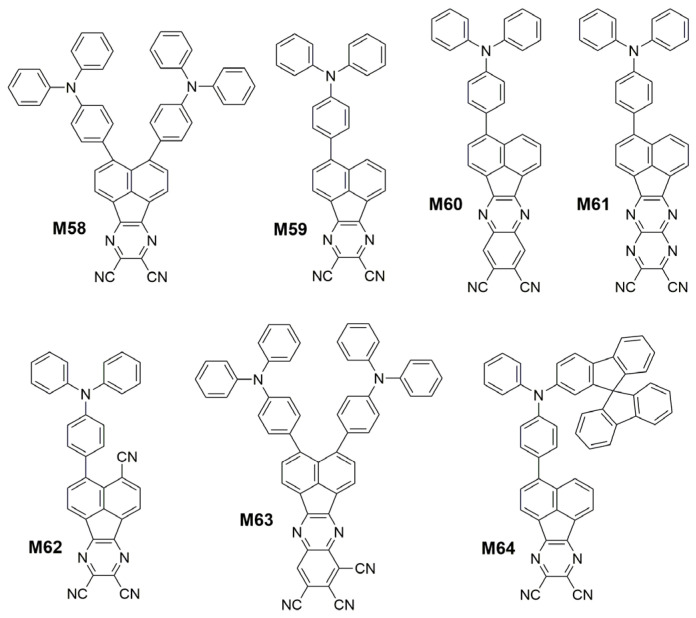
Chemical structures of **M58** to **M64**.

**Figure 20 polymers-15-00098-f020:**
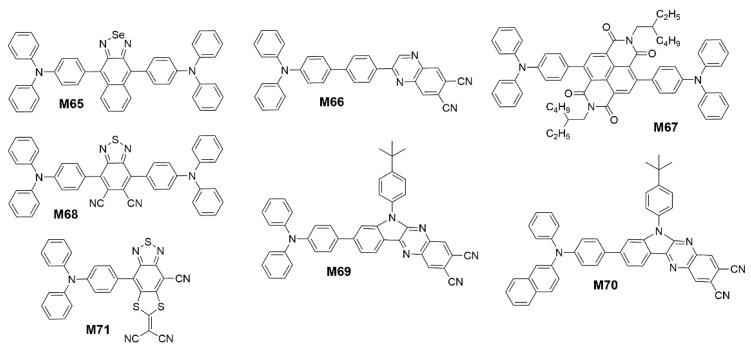
Chemical structures of **M65** to **M71**.

**Figure 21 polymers-15-00098-f021:**
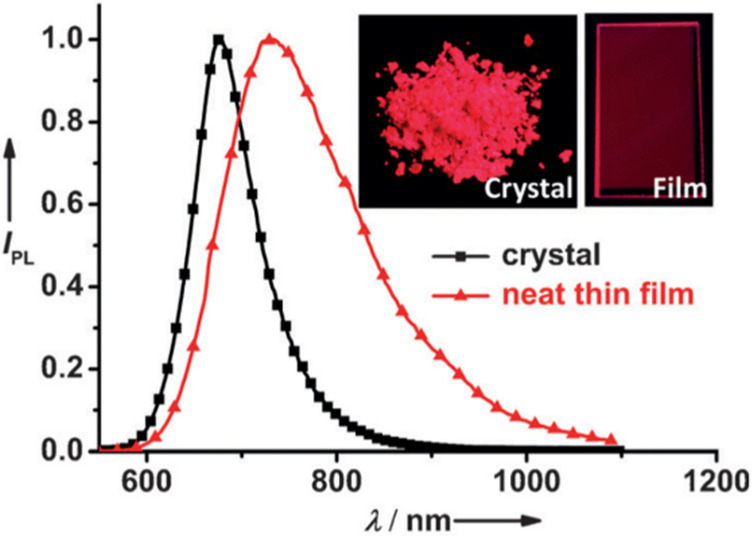
PL spectra of **M66** crystal and neat thin film at room temperature. Inset: images of crystal and neat thin film under UV irradiation (λ_ex_ = 365 nm) (adapted from ref. [165]).

**Figure 22 polymers-15-00098-f022:**
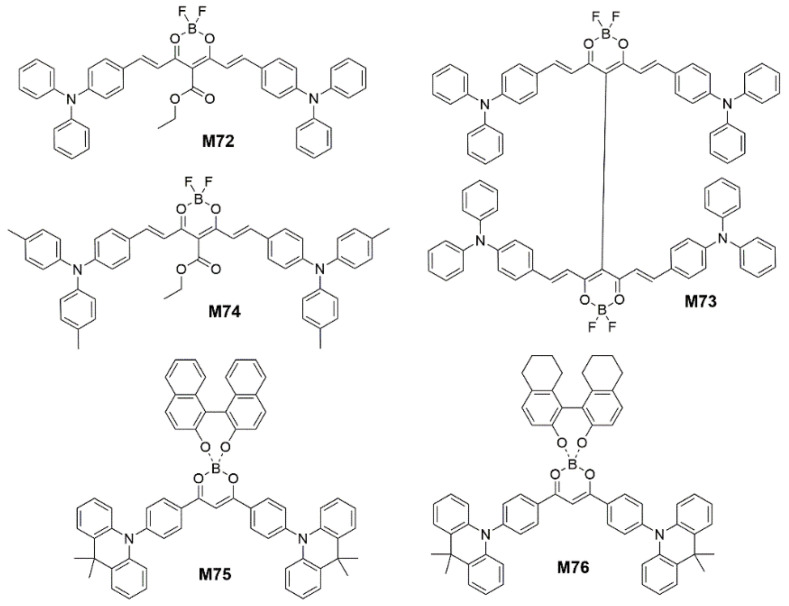
Chemical structures of **M72** to **M76**.

**Figure 23 polymers-15-00098-f023:**
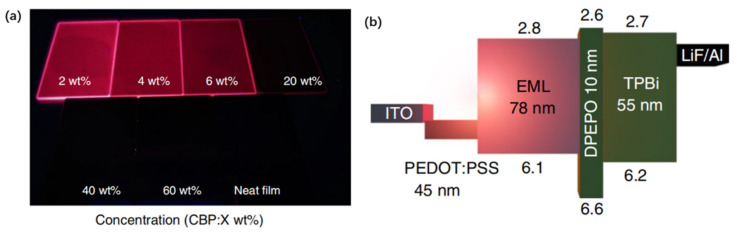
(**a**) Photographs under UV light (each sample: 1.5 × 2 cm^2^) of **M72**; (**b**) Schematic representation and energy diagram of the NIR OLEDs (adapted from ref. [171]).

**Table 1 polymers-15-00098-t001:** Photophysical and electroluminescent properties of some NIR conjugated polymers.

	λ_PL_ (nm)	*Φ*_P_ (%)	λ_EL_ (nm)	*EQE*_max_ (%)	Ref.
**P1**	517, 554 (shoulder)	0.17	690	-	[78]
**P2**	652	0.07	800	-	[78]
**P3**	-	-	705	0.005	[79]
**P4**	-	-	950	0.0005	[79]
**P5**	-	-	956 (unblended) 876 (blended)	0.003 (unblended) 0.002 (blended)	[41]
**P6**	730	-	742	0.3	[82]
**P7**	418, 881	-	909	0.04	[83]
**P8**	880	-	885	0.27	[84]
**P9**	650, 945	-	930	0.004	[85]
**P10**	697	15	697	0.70	[40]
**P11**	790	1	790	0.02	[40]
**P12**	890	-	895	0.09	[86]
**P13**	1000	-	990	0.02	[86]
**P14**	718	-	708	0.69	[87]
**P15**	527	-	-	-	[42]
**P16**	731 (solid)	23	716	0.41	[42]
**P17**	750	-	758	2.12	[90]
**P18**	641 (solution) 648, 776 (film)	26.32	772	0.38	[91]
**P19**	655 (solution) 698, 810 (film)	28.53	764	0.17	[91]
**P20**	642 (solution) 657, 766 (film)	31.98	775	0.63	[91]

**Table 2 polymers-15-00098-t002:** Photophysical and electroluminescent properties of some NIR fluorescent organic molecules.

	λ_PL_ (nm)	*Φ*_P_ (%)	λ_EL_ (nm)	*EQE*_max_ (%)	Ref.
**M1**	595, 648 (shoulder)	-	720, 805 (shoulder)	0.0018	[93]
**M2**	922 (pellet)	-	936	-	[95]
**M3**	905 (film)	-	910	-	[96]
**M4**	945 (pellet)	-	891	-	[97]
**M5**	-	-	800	1.9	[98]
**M6**	636, 687 (shoulder)	-	848	0.0012	[99]
**M7**	975	7.4	1050	0.05	[103]
**M8**	1120	4.9	1115	-	[103]
**M9**	975, 1050 (film)	7.4	1050	0.16	[104]
**M10**	995, 1060 (film)	5.8	1080	0.73	[104]
**M11**	970, 1040 (film)	6.3	1050	0.33	[104]
**M12**	1255	0.5	1220	-	[104]
**M13**	784	5.3	752 (doped) 784 (nondoped)	1.12 (doped) 0.02 (nondoped)	[105]
**M14**	800	4.4	748 (doped) 788 (nondoped)	1.13 (doped) 0.03(nondoped)	[105]
**M15**	868	4.0	823 (doped) 870 (nondoped)	0.27 (doped) 0.02 (nondoped)	[105]
**M16**	698	21	692	1.6	[106]
**M17**	805	7.6	815	0.5	[106]
**M18**	725	0.07	730	0.28	[107]
**M19**	700, 704 (film)	10.1	706	0.89	[43]
**M20**	780, 761 (film)	0.28	749	0.29	[43]
**M21**	787, 803 (film)	13.0	802	0.43	[43]
**M22**	857, 883 (film)	0.20	864	0.20	[43]
**M23**	712	-	688	3.13	[113]
**M24**	700, 725 (film)	15, 34 (film)	726	1.51	[118]
**M25**	702, 725 (film)	13, 10 (film)	726	0.96	[118]
**M26**	761, 760 (film)	16, 22 (film)	754	1.48	[119]
**M27**	661, 720 (film)	10, 0.8 (film)	734	0.48	[120]
**M28**	686, 758 (film)	5, 0.6 (film)	773	0.26	[120]
**M29**	-	-	640	0.054	[121]
**M30**	-	-	700	0.051	[121]
**M31**	728	86	692 (doped) 708 (nondoped)	3.8 (doped) 3.1 (nondoped)	[122]
**M32**	668	25.2	683	0.57	[123]
**M33**	840 (doped)	19	840 (doped)	1.01	[124]
**M34**	702	10	679	0.48	[127]
**M35**	716	9.5	716	0.4	[128]
**M36**	700	20	720	1.1	[129]
**M37**	658	-	756	1.87	[130]
**M38**	785	6, 11 (film)	786	0.77	[131]
**M39**	746	26	774	5.3	[132]

**Table 3 polymers-15-00098-t003:** Photophysical and electroluminescent properties of some NIR phosphorescent organic molecules.

	λ_PL_ (nm)	*Φ*_P_ (%)	λ_EL_ (nm)	*EQE*_max_ (%)	Ref.
**M40**	670	52	700	2.1	[136]
**M41**	660	-	692	2.4	[49]
**M42**	700 (film)	16	700	1.54	[140]
**M43**	683	60	696	3.9	[142]
**M44**	642	-	738	0.82	[143]
**M45**	683 (film)	30	700	1.47	[144]
**M46**	710 (film)	17	702	1.2	[145]
**M47**	736, 780 (film)	26	718	1.58	[147]
**M48**	714, 732 (film)	38	693	1.33	[147]
**M49**	724, 748 (film)	34	707	3.02	[147]
**M50**	708 (film)	14	710	2.1	[150]
**M51**	709	48.84	718	5.4	[38]
**M52**	692 (film)	13	716	1.9	[152]
**M53**	710 (film)	7	748	1.4	[152]
**M54**	702 (film)	11	734	1.4	[152]
**M55**	702 (film)	6	748	1.0	[152]
**M56**	735	39 (doped)	698	7.68	[153]
**M57**	729	19.1	734	13.4	[154]
**M58**	756 (film)	17	777	2.19	[155]
**M59**	777	20.3	765	5.1	[158]
**M60**	716	16.3	711	3.5	[158]
**M61**	742 (solution) 1009 (film)	7.7 (solution)	722 (doped) 1010 (nondoped)	1.35 (doped) 0.003 (nondoped)	[159]
**M6** **2**	887 (film)	0.18	904	0.019	[160]
**M6** **3**	878 (film)	1.10	910	0.22	[161]
**M6** **4**	758 (film)	15	782	2.55	[162]
**M6** **5**	764 (film)	12	730	2.65	[163]
**M6** **6**	733 (film)	21	728	3.9	[164]
**M6** **7**	740	8.4	740	2.4	[165]
**M6** **8**	750 (film)	21	712	6.57	[166]
**M6** **9**	696	35.8	709	6.61	[167]
**M** **70**	696	64.4	711	9.44	[167]
**M** **71**	820	10.7	750 (doped) 802 (nondoped)	14.3 (doped) 2.2(nondoped)	[39]
**M** **72**	721 (doped) 782 (film)	0.7 (doped) 0.035 (film)	721 (doped)	9.74 (doped)	[168]
**M** **73**	760 (doped) 788 (solution)	45 (doped) 1.3 (solution)	758	5.1	[169]
**M** **74**	840 (doped)	15.8 (doped)	840 (doped)	3.8 (doped)	[170]
**M** **75**	536	1	716	1.9	[171]
**M** **76**	534	2	700	0.7	[171]

## Data Availability

Not applicable.

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
