# Peer review of "Progresses and Perspectives of Near-Infrared Emission Materials with “Heavy Metal-Free” Organic Compounds for Electroluminescence"

_polymers, 2022, doi:10.3390/polym15010098_

Round 1

Reviewer 1 Report

Here, authors review progresses in conjugated polymers and organic molecules for NIR OLEDs/PLEDs applications.

In this reviewer opinion the topic is interesting although some paragraphs are very confusing.

-“In the typical π-conjugated system”; which ones are typical?

-“Hybridization of the energy levels of the donor and acceptor could rise the energy level of the HOMO higher than that of donor and reduce the energy level of LUMO than that of acceptor, leading to an unusually small HOMO-LUMO separation” This is unreadable.

 -“The smaller bond length alternation is, the lower energy gap of a conjugated compound will be. As a result, shortening the bond length alternation is a significant step toward the reduction of the energy gap in the conjugated systems.” What means shortening the bond length alternation?

In addition, the length of the manuscript should be reduced pointing to the most interesting things to make reading easier. Potential readers can later go to the reference for further information.

Some additional comments are listed below.

-Authors write many times OLEDs/PLEDs even if they are writing about organic molecules or polymers. In my opinion, authors should refer to OLEDs with organic systems and PLEDs when authors write about conjugate polymers. For instance, in section 3 NIR fluorescent materials based on polymers, authors use along the section OLEDs/PLEDs.

-Line 103 from probably???

-Line 577 meaned should be meant

-Lines 760-761: “   traditional organic near-infrared fluorescent OLEDs (FOLEDs) is …” Authors can easily find that the acronym FOLEDs refers to flexible organic LEDs.

-Along the text, there are many sentences two long and/or unreadable. For instance:

“In 2015, Li et al. [49] proposed an open-shell organic molecule M41 as an NIR-emitter of OLEDs which could circumvent the transition problem of triplet through the doublet theory that there was only one unpaired electron in the highest SOMO.”

Line 63:  “Greatly improve quantum efficiency effectively” I think authors will find a better way to construct the sentences.

Line 205: authors need to correct the units of the atomic radius to 102 pm.

Line 300: Instead of “In the next year,   “ I would rather recommend “One year later” or “In 2009,”

Line 356. Replace “And in 2020,” by In 2020,

446-447: In 2018, Yang et al. [131] realized triplet-singlet energy transfer by the Förster mechanism. This sentence is not readable. Authors should take care about the meaning of the verb “realize”.

Line 67: “The materials with doublet …” Authors need to clarify what they refer to.

Overall, authors should reread the whole manuscript trying to reduce the length of the manuscript and of the too long sentences as that complicates the reading. Sentences like (lines 111-112) “…the energy gap is generated by alternating bond length between the single bond and the double bond ….”. can be said “the energy gap is generated by alternating single and double bonds” significantly reducing the length of the sentences and thus making the reading easier.

Author Response

Response to Reviewer 1 Comments

Here, authors review progresses in conjugated polymers and organic molecules for NIR OLEDs/PLEDs applications.

In this reviewer opinion the topic is interesting although some paragraphs are very confusing.

Point 1

-“In the typical π-conjugated system”; which ones are typical?

Response 1

The typical π-conjugated system refers to the polycyclic aromatic hydrocarbons mentioned above. We have changed the words in the revised version.

Point 2

-“Hybridization of the energy levels of the donor and acceptor could rise the energy level of the HOMO higher than that of donor and reduce the energy level of LUMO than that of acceptor, leading to an unusually small HOMO-LUMO separation” This is unreadable.

Response 2

We have changed the expression of this sentence in the revised version.

Point 3

 -“The smaller bond length alternation is, the lower energy gap of a conjugated compound will be. As a result, shortening the bond length alternation is a significant step toward the reduction of the energy gap in the conjugated systems.” What means shortening the bond length alternation?

Response 3

We have used "reducing" instead of "shortening" in the revised version.

Point 4

In addition, the length of the manuscript should be reduced pointing to the most interesting things to make reading easier. Potential readers can later go to the reference for further information.

Some additional comments are listed below.

-Authors write many times OLEDs/PLEDs even if they are writing about organic molecules or polymers. In my opinion, authors should refer to OLEDs with organic systems and PLEDs when authors write about conjugate polymers. For instance, in section 3 NIR fluorescent materials based on polymers, authors use along the section OLEDs/PLEDs.

Response 4

We have used the “OLEDs” with organic systems and the “PLEDs” with polymers rather than “OLEDs/PLEDs” in the revised version.

Point 5

-Line 103 from probably???

Response 5

The sentence of Line 103 is from reference [65].

Point 6

-Line 577 meaned should be meant

Response 6

We have corrected the mistake in the revised version.

Point 7

-Lines 760-761: “  traditional organic near-infrared fluorescent OLEDs (FOLEDs) is …” Authors can easily find that the acronym FOLEDs refers to flexible organic LEDs.

Response 7

We have deleted the word “(FOLEDs)” in the revised version.

Point 8

-Along the text, there are many sentences two long and/or unreadable. For instance:

“In 2015, Li et al. [49] proposed an open-shell organic molecule M41 as an NIR-emitter of OLEDs which could circumvent the transition problem of triplet through the doublet theory that there was only one unpaired electron in the highest SOMO.”

Response 8

Based on the comments, the whole paper was checked to avoid the long or unreadable sentences. And we have changed the expression of this pointed sentence in the revised version.

Point 9

Line 63: “Greatly improve quantum efficiency effectively” I think authors will find a better way to construct the sentences.

Response 9

We have changed the expression of the sentence in the revised version.

Point 10

Line 205: authors need to correct the units of the atomic radius to 102 pm.

Response 10

We have corrected the mistake in the revised version.

Point 11

Line 300: Instead of “In the next year, “ I would rather recommend “One year later” or “In 2009,”

Response 11

We have changed the words in the revised version.

Point 12

Line 356. Replace “And in 2020,” by In 2020,

Response 12

We have changed the words in the revised version.

Point 13

446-447: In 2018, Yang et al. [131] realized triplet-singlet energy transfer by the Förster mechanism. This sentence is not readable. Authors should take care about the meaning of the verb “realize”.

Response 13

We have used "achieved" instead of " realized " in the revised version.

Point 14

Line 67: “The materials with doublet …” Authors need to clarify what they refer to.

Response 14

We have changed the expression of the sentence in the revised version.

Point 15

Overall, authors should reread the whole manuscript trying to reduce the length of the manuscript and of the too long sentences as that complicates the reading. Sentences like (lines 111-112) “…the energy gap is generated by alternating bond length between the single bond and the double bond ….”. can be said “the energy gap is generated by alternating single and double bonds” significantly reducing the length of the sentences and thus making the reading easier.

Response 15

Based on the comments, the whole paper was checked to avoid the long or unreadable sentences. And we have changed the expression of the pointed sentence in the revised version.

Author Response

Response to Reviewer 2 Comments

The authors reviewed “heavy metal-free” organic/polymer semiconducting materials for near-infrared light emitting. Plenty of polymers and molecules were listed and detailed discussions were provided with valuable information. The review briefly introduced the mechanisms of electroluminescence and strategies on tuning the Eg to NIR region, followed by summary of semiconducting polymers and molecules. It would be better to enrich the introduction/physics part, otherwise this work looks more like a list of materials.

Point 1

What’s the advantages of organic materials over the “heavy metal” related ones? Please add more explanations in terms of physics rather than cost only.

Response 1

In addition to the cost, the mechanical adaptability of organic NIR light-emitting materials also makes them have broad application prospects in flexible and stretchable devices. In addition, metal-free organic light-emitting materials could be used as biocompatible substitutes for inorganic materials, and could be used in implantable, wearable or medical applications in vivo. We have added the point in the revised version.

Point 2

In the mechanism part, what’s the main one for the NIR OLEDs with high efficiency. How to improve the performance of NIR OLEDs?

Response 2

Improving the exciton utilization is important factor to improve the luminous efficiency of devices. In addition, the inhibition of non-radiative transition caused by molecular vibration relaxation and the inhibition of intermolecular accumulation to reduce the quenching effect are helpful to improve the efficiency of OLED.

Point 3

It would be better to provide a sample with NIR emission in Figure 4.

Response 3

It's a pity that we can't get the copyright of the Figure of the reference [70]. Moreover, the molecules listed in Fig. 4b are intended to represent the D-A effect that redshifts the wavelengths of luminescence, but they have not yet redshifted to the near-infrared region.

Point 4

The full name of polymers/molecules should be mentioned in the text before labeling them as Pxx or Mxx.

Response 4

We have added the full name of polymers/molecules as far as possible in the revised version.

Point 5

What’s the challenge or opportunity of NIR OLEDs?

Response 5

After continuous development in recent years, NIR luminescent materials have achieved some phased results. The quantum yield of the material, the electroluminescence wavelength, stability and EQE of the device have been greatly improved. However, the types of high-performance organic NIR luminescence materials are still few, and the problems of low device efficiency and efficiency roll-down are still the bottlenecks in the development and application. The future directions may include the following fields: 1) develop new and efficient organic near infrared luminescence materials; 2) study the relationship between molecular structure and electroluminescence properties; 3) reveal the relationship between device properties and device processing technology, and host and guest energy levels; 4) realize the application of organic NIR luminescence materials and devices in military, optical fiber communication, biomedical imaging and other fields.

We have added these comments in the revised version.

Round 2

Reviewer 1 Report

Most questions/comments made by this reviewer have been addressed by the authors.

However, I could not find the sentence in reference 65 as authors responded:

"The sentence of Line 103 is from reference [65]".

That sentece needs to be clarified.

Author Response

Response to Reviewer 1 Comments

Point 1

Most questions/comments made by this reviewer have been addressed by the authors.

However, I could not find the sentence in reference 65 as authors responded:

"The sentence of Line 103 is from reference [65]".

That sentence needs to be clarified.

Response

Reference 65, page 1403, line 7 from the bottom on the left, said "the absorption peak of 7 is at 877nm, and that of 8 is at 953nm." "8" in the reference was the molecule "n=4" in the Figure 2 of this paper. And the "577 nm" was the absorption peak of the molecule "n=0" calculated proportionally according to the spectrogram in Figure 2.

Reviewer 2 Report

The authors addressed most of my concerns. Please double check the figure numbers after revision (e.g. Figure 25) and make sure every figure has been mentioned in the main text.

Author Response

Response to Reviewer 2 Comments

Point 1

The authors addressed most of my concerns. Please double check the figure numbers after revision (e.g. Figure 25) and make sure every figure has been mentioned in the main text.

Response

Thanks for your careful examination. The serial number of Figure 25 is wrong, it should be Figure 23. We have corrected this error and double check the figure numbers in the revised version.
